# How to Break Image Classification Models: Random Noise knows better than Saliency

## Abstract

In safety critical domains such as autonomous driving or medical diagnosis, it is not enough to know what image classifiers focus on, we must also understand how vulnerable their decisions are to the removal of supposedly important evidence. We propose a degradation-based evaluation framework built on the central hypothesis that masking the most important input regions should cause maximal prediction collapse. Attribution methods such as SHAP, Grad-CAM, or Integrated Gradients are therefore treated as perturbation guides, and their effectiveness is measured by the extent to which they accelerate model degradation. Our framework produces degradation curves from which we derive three quantitative metrics: Area Under the Blindness Curve (AUBC), Sensitivity Slope, and Attribution Collapse Point. This shifts evaluation from judging saliency maps by visual plausibility to assessing their behavioral impact on model predictions. Strikingly, across standard architectures we find that random masking often surpasses attribution based masking, revealing a fundamental gap between visual explanations and true decision dependencies. By exposing this gap, our framework enables reproducible, model agnostic robustness analysis and opens the path toward vision models that are empirically aware of their own failure modes.

## 1 Introduction

Modern image classifiers are increasingly deployed in safety-critical domains such as autonomous driving Pei et al. (2017); Geiger et al. (2012); Eberhardt et al. (2024); Rosenfeld et al. (2018) and medical imaging DeGrave et al. (2020); Lin et al. (2019); Hou et al. (2024). In these settings, accuracy alone is insufficient: we also need systematic ways to probe when models are vulnerable and how their predictions collapse under stress. This motivates a shift from purely interpretive explanations toward evaluation frameworks that connect attribution with robustness. We start from a simple hypothesis: if an attribution map highlights the regions a model truly relies on, then masking those regions should lead to maximal prediction degradation. Building on this idea, we introduce a degradation-based evaluation framework that progressively removes input evidence, guided by attribution methods such as SHAP Lundberg & Lee (2017), Grad-CAM Selvaraju et al. (2019), and Integrated Gradients Sundararajan et al. (2017). The resulting degradation curves quantify how confidence declines as regions are masked, and from these curves we derive three metrics: Area Under the Blindness Curve (AUBC), Sensitivity Slope, and Attribution Collapse Point (ACP). Figure 1 illustrates example saliency maps for the Vigo (dog) image.

Our experiments, however, reveal a surprising outcome: randomly guided masking often produces stronger degradation than attribution guided masking. This does not imply that attribution methods are invalid, but rather that degradation is not a straightforward proxy for attribution fidelity. Instead, degradation curves expose model specific vulnerabilities and provide a quantitative lens on robustness, independent of visual plausibility. In summary, our contribution is to present a degradation based evaluation framework that formalizes the link between attribution and robustness, introduces three reproducible metrics for quantifying prediction collapse, and demonstrates that degradation analysis uncovers unexpected vulnerabilities in standard classifiers. Rather than judging attribution methods, our framework establishes degradation as a principled tool for robustness diagnosis and for developing models that can recognize their own failure modes.

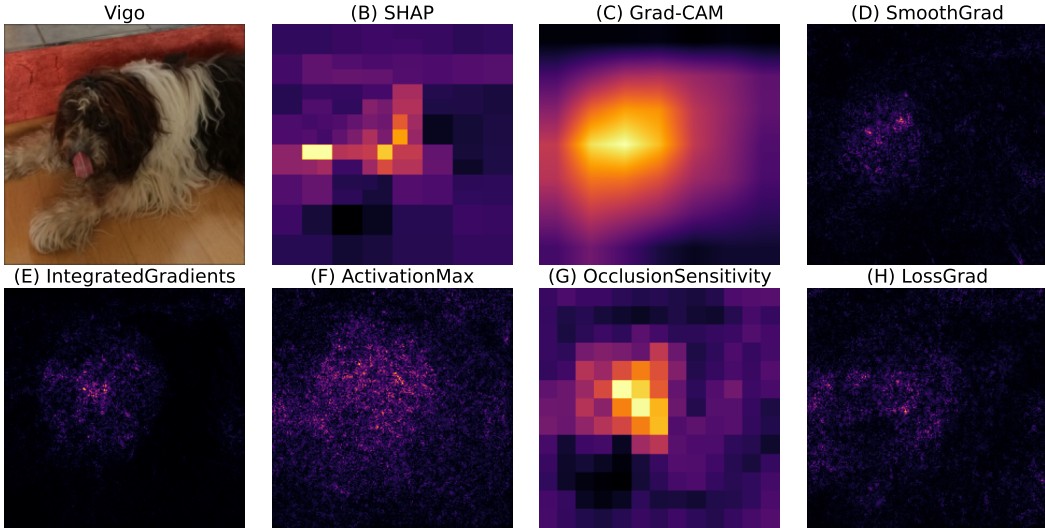

Figure 1: Saliency maps for "Vigo" (dog) image generated using different explanation methods (normalized to $[0, 1]$ and resized to input resolution).

## 2 RELATED WORK

To evaluate whether attribution methods can expose true model dependencies, we draw on four areas of prior work: **saliency methods** as guidance tools, **evaluation strategies** beyond visual plausibility, **interpretability taxonomies** that frame our functional approach, and **causal reasoning** as intervention based justification. Together, these foundations support our goal to systematically break models.

**Saliency and Attribution Methods.** Saliency methods aim to identify the input regions most responsible for a model's prediction. Early gradient based techniques such as Saliency Maps Simonyan et al. (2014), Deconvolution Zeiler & Fergus (2013), and Guided Backpropagation Springenberg et al. (2015) visualize gradient information, but later studies showed that some outputs may ignore model weights entirely. Our framework tests such methods not by plausibility, but by their ability to reduce model confidence when salient regions are progressively masked. CAM and Grad-CAM Zhou et al. (2015); Selvaraju et al. (2019), along with their extensions like Grad-CAM++ Chattopadhay et al. (2018) and Score-CAM Wang et al. (2020b), leverage spatial activations to produce localized heatmaps. However, their behavioral impact remains underexplored. Integrated Gradients Sundararajan et al. (2017) and DeepLIFT Shrikumar et al. (2019) offer axiomatic relevance attributions, while LRP Bach et al. (2015) and SHAP Lundberg & Lee (2017); Shapley & Corporation (1951) use propagation and game theoretic formulations, respectively. Rather than evaluating these methods by fidelity alone, we ask: can they break the model when used as masking guides? Methods like SmoothGrad and RISE Wang et al. (2020a); Petsiuk et al. (2018) introduce stochasticity or random masking, conceptually aligned with our intervention based view. LIME and Meaningful Perturbations Ribeiro et al. (2016); Fong & Vedaldi (2017) treat attribution as counterfactual intervention but lack scalability. Our method builds on these ideas by offering consistent, model agnostic metrics that directly quantify degradation effects across architectures.

**Evaluation and Faithfulness.** Concerns about saliency reliability have spurred quantitative evaluations. Sanity checks Adebayo et al. (2020) showed that several methods produce similar outputs even when model weights are randomized. We extend this line of critique with targeted degradation: attribution maps are used not for visualization but to erode the model's confidence. ROAR Hooker et al. (2019) assesses saliency by removing input regions and retraining the model—our method achieves similar evaluation without retraining by tracking confidence decay in a single forward pass pipeline. Deletion/insertion scores Petsiuk et al. (2018) and infidelity metrics Yeh et al. (2019) capture similar principles, but lack the granularity and continuity of our degradation curves. Adversarial studies Ghorbani et al. (2018); Dombrowski et al. (2019) reveal fragility through imper-

ceptible perturbations. We instead apply structured, semantically meaningful masking that reveals the breakdown threshold. Our approach also aligns with recent calls for scalable, standardized evaluation protocols Duan et al. (2024), offering architecture independent metrics like AUBC, sensitivity slope, and collapse point to benchmark attribution performance.

**Interpretability Taxonomies and Evaluation Criteria.** Doshi-Velez and Kim Doshi-Velez & Kim (2017) classify interpretability evaluations into application grounded, human grounded, and functionally grounded. Our method is explicitly functionally grounded: it evaluates explanation methods based on their behavioral effect on the model. Jacovi and Goldberg Jacovi & Goldberg (2020) highlight the disconnect between faithfulness and visual plausibility, cautioning against relying solely on human perceived coherence. We empirically support this distinction by showing that visually plausible maps often fail to degrade model predictions—and may underperform even random masking. Mohseni et al. Kadir et al. (2023) offer a comprehensive taxonomy of XAI metrics, spanning fidelity, robustness, consistency, and usability. Our degradation framework specifically addresses fidelity and robustness through behavioral intervention rather than perceptual coherence.

**Causal Inference and Robust Reasoning.** Several works frame attribution as a form of causal inference. Narendra et al. Narendra et al. (2018) and Chattopadhyay et al. Chattopadhyay et al. (2019) estimate feature effects via structural causal models, while SCOUT Wang & Vasconcelos (2020) generates causal heatmaps based on learned assumptions. Diffusion counterfactuals Augustin et al. (2022) aim for realism but are computationally expensive. Although we share the interventionist spirit of these approaches, we do not rely on causal modeling. Instead, we assess explanatory strength via empirical degradation: if masking the most attributed regions does not break the model, then those attributions lack behavioral grounding. Related evaluation metrics like sensitivity Yeh et al. (2019) and deletion scores Petsiuk et al. (2018) are special cases of our framework, which generalizes such evaluations through continuous degradation curves and interpretable collapse dynamics. Attribution fragility also links to adversarial robustness: adversarial training has been shown to improve attribution quality Etmann et al. (2019); Ilyas et al. (2019), and shape biased models exhibit more aligned saliency maps Geirhos et al. (2022). Our metrics capture such shifts in attribution robustness, and inspired by AutoAttack Athalye et al. (2018), we propose a fully automated, parameter free evaluation pipeline.

**Research Gap and Our Contribution.** While saliency methods are widely used to interpret model predictions, their ability to identify functionally critical features remains largely untested. Existing evaluations focus on visual plausibility or basic sanity checks, without examining whether explanations provoke measurable prediction collapse. We address this gap by introducing a behavioral evaluation framework that uses attribution as a perturbation guide to break models. This yields degradation curves from which we derive three robustness metrics. Our results show that popular attribution methods often underperform even random masking highlighting a disconnect between visual coherence and true decision relevance.

## 3 How to Break - Methodology

This section introduces our framework for attribution aware model degradation. We investigate the prediction sensitivity of image classification models by progressively perturbing input regions. We formalize degradation curves, describe how they are interpolated, and derive robustness metrics that quantify how easily a model's predictions collapse. In contrast to prior work, we do not assume that attribution maps are inherently faithful, we empirically test their alignment with model behavior.

### 3.1 Model Degradation

We define attribution guided vulnerability as the extent to which a model's prediction empirically depends on specific regions of the input, such that perturbing these regions leads to significant changes in output confidence. Formally, let $f : \mathcal{X} \to \mathcal{Y}$ be a trained classifier, and let $x \in \mathcal{X}$ be an input with true label $y \in \mathcal{Y}$. A perturbation operator $\delta : \mathcal{X} \times \mathcal{M} \to \mathcal{X}$ modifies the input based on a binary or real valued mask $M$, producing a degraded version $x' = \delta(x, M)$ that omits or alters specific regions. We interpret $x'$ as a hypothetical variation of $x$, a localized input modification that enables empirical estimation of feature relevance. If the model's confidence in class $y$ drops significantly under $\delta(x, M)$, we infer that the affected region contributed meaningfully to the prediction.

This setup is aligned with interventionist evaluation, where the effect of a variable is assessed by observing changes under manipulation.

However, our framework is purely observational and model centric: it does not assume access to structural causal models or counterfactual ground truth. We define the empirical influence of a masked region $S \subseteq 1, \ldots, H \times W$ as:

$$\mathrm{EI}_f(S \mid x) = f_y(x) - f_y(\delta(x, M_S)), \tag{1}$$

where $M_S$ is a mask that removes or replaces the pixels in $S$. Higher influence scores indicate stronger model sensitivity to the region $S$. By ranking and perturbing regions according to a given guidance strategy, we estimate the cumulative impact of input suppression on model predictions.

This allows us to treat the degradation curve $D(t)$ as a trajectory of prediction sensitivity under structured perturbation—not as a causal quantity, but as an empirical diagnostic of how a model depends on different regions of its input. This, in turn, enables comparisons between attribution methods by evaluating how effectively they prioritize prediction critical features. This perspective aligns with the view of Jacovi and Goldberg Jacovi & Goldberg (2020), who argue that explanations should be assessed not merely by visual plausibility but by their behavioral faithfulness to the model's internal decision process. We emphasize that our protocol evaluates behavioral sensitivity under constrained, standardized perturbations and does not aim to estimate causal effects.

## 3.2 FRAMEWORK OVERVIEW

We aim to evaluate how sensitive image classifiers are to targeted, guided perturbations. Unlike adversarial attacks that maximize loss, our method progressively removes input regions based on different guidance strategies, thereby revealing how predictions collapse as evidence is withdrawn. Our framework consists of two key components:

**1. Guidance.** We define a function $g : \mathbb{R}^{3 \times H \times W} \to [0, 1]^{H \times W}$ that assigns an importance score to each pixel location. This guidance map can be generated using attribution methods (e.g., SHAP, Grad-CAM, Integrated Gradients) or by a random baseline. The values in $g$ determine the order in which pixels or regions are selected for perturbation.

**2. Constrained Perturbation Evaluation.** Given a fixed guidance map $g$ and a perturbation operator $\delta : \mathbb{R}^{3 \times H \times W} \times \mathcal{M} \to \mathbb{R}^{3 \times H \times W}$, where $\mathcal{M}$ is a binary mask with $M \in 0, 1^{H \times W}$, we iteratively mask regions of the input image $x$ by applying $\delta$ to the highest ranked pixels according to $g$. At each perturbation step $t$, we measure the change in model confidence for the ground truth class $y$ using the predictive output $f : \mathbb{R}^{3 \times H \times W} \to [0, 1]^C$.

## 3.3 GUIDANCE STRATEGIES

We investigate a range of strategies that can guide region selection for progressive input perturbation. Attribution methods such as SHAP Lundberg & Lee (2017), Grad-CAM Selvaraju et al. (2019), and Integrated Gradients Sundararajan et al. (2017) produce saliency maps that highlight regions deemed important by the model. Random masking serves as a control condition, removing regions in a uniform random order without reference to model specific evidence. SHAP Lundberg & Lee (2017) estimates Shapley values from coalitions of input features and is grounded in cooperative game theory. Grad-CAM Selvaraju et al. (2019) generates relevance maps by computing gradient-weighted class activation mappings, offering class specific localization. SmoothGrad Wang et al. (2020a) enhances gradient based explanations by averaging noisy gradients to highlight stable patterns. Integrated Gradients Sundararajan et al. (2017) accumulate gradients along a linear path from a baseline to the input, capturing attribution contributions more robustly. Activation Maximization Zhou et al. (2018) identifies patterns that strongly activate target neurons, while loss gradient methods Zhou et al. (2018); Goodfellow et al. (2015); Madry et al. (2019) compute the gradient of the loss with respect to the input. In contrast to attribution derived guidance, Occlusion Sensitivity directly perturbs local patches and measures the resulting change in prediction confidence, providing a ground truth inspired baseline of influence.

## 3.4 DEGRADATION CURVE INTERPOLATION

To characterize how models respond to progressive input removal, we approximate the degradation trajectory $D(r)$ with a smooth parametric function. Specifically, we use a three parameter sigmoid of the form:

$$\hat{D}(r) = a \left( 1 - \frac{1}{1 + e^{-b(r-c)}} \right), \tag{2}$$

where $r \in [0, 1]$ denotes the cumulative input blindness (mask ratio), and $a$, $b$, and $c$ control the drop magnitude, steepness, and inflection point, respectively. The parameters are estimated via nonlinear least squares using SciPy's `curve_fit` routine Virtanen et al. (2020). To improve robustness against noise and outliers, we apply envelope binning: the blindness-confidence data $(r_i, D(r_i))_{i=1}^{N}$ is grouped along the $r$-axis, and within each bin a high quantile (here: 99.999%) is extracted to form a smoothed upper envelope. This filtered trajectory is then used for sigmoid fitting, ensuring that $\hat{D}(r)$ captures the dominant degradation trend while suppressing small perturbation artifacts.

## 4 HOW TO MEASURE – ROBUSTNESS EVALUATION METRICS

From the degradation curve $D(r)$, obtained via progressive, guidance driven perturbation, we derive three metrics that quantify a model's robustness to structured input removal. These metrics capture how rapidly, severely, and irreversibly model confidence declines as evidence is withdrawn. Each point on $D(r)$ corresponds to a perturbation $\delta(x, M_r)$, where a fraction $r$ of the input is suppressed. The value $D(r) = f_y(\delta(x, M_r))$ reflects the model's confidence after masking region $S_r$, serving as an empirical measure of prediction stability. To assess both global and object specific degradation, we distinguish two masking types. Absolute Blindness $r^{\text{abs}}$ is the fraction of the full image masked, while Relative (Object) Blindness $r^{\text{rel}}$ refers to the masked proportion of a segmented object region $S$. Both are computed as:

$$r_t^{\text{abs}} = \frac{\|1 - M_t\|_1}{H \cdot W} \qquad r_t^{\text{rel}} = \frac{\|(1 - M_t) \cdot S\|_1}{\|S\|_1}, \tag{3}$$

where $M_t$ is the binary mask at perturbation step $t$, and $S$ is the binary mask denoting the object region. While $r^{\text{abs}}$ captures the overall degree of input perturbation, $r^{\text{rel}}$ quantifies the proportion of the object that has been blinded. This dual formulation allows us to evaluate degradation behavior both in terms of total input coverage and semantic object relevance.

### 4.1 AREA UNDER THE BLINDNESS CURVE (AUBC)

We define the Area Under the Blindness Curve (AUBC) as the integral over the model's confidence trajectory with respect to the blindness ratio $r_i \in [0, 1]$. AUBC summarizes the total retained confidence along the degradation path. Formally, it is computed as:

$$\text{AUBC} = \sum_{i=1}^{K} \frac{(r_i - r_{i-1})(D_i + D_{i-1})}{2}, \tag{4}$$

where $r_i$ may represent either the absolute blindness $r^{\text{abs}}$ or the object relative blindness $r^{\text{rel}}$, depending on the analysis context. The term $D_i = f_y(\delta(x, M_{r_i}))$ denotes the model's predicted confidence for the ground truth class at masking level $r_i$. A higher AUBC value indicates that the model retains confidence even as large portions of the input are suppressed, suggesting robustness to input degradation. Conversely, a lower AUBC implies greater prediction fragility and a stronger empirical dependence on the removed regions.

### 4.2 SENSITIVITY SLOPE

The sensitivity slope $\nabla C$ captures the model's peak vulnerability—how abruptly its confidence declines when the most prediction critical regions are removed. Let $\hat{D}(r)$ denote the sigmoid fit to the degradation curve. The slope is defined as:

$$\nabla C = \max_{r \in [0,1]} \left| \frac{d\hat{D}(r)}{dr} \right|. \tag{5}$$

A steep slope indicates that the model relies heavily on a small subset of features, making it sensitive to localized perturbations. In contrast, a flatter curve suggests greater robustness.

### 4.3 ATTRIBUTION COLLAPSE POINT (ACP)

The Attribution Collapse Point (ACP) denotes the blindness level at which the model's confidence first drops below a threshold $\tau$ (e.g., $\tau = 0.5$). It identifies the tipping point at which the prediction becomes unreliable:

$$\text{ACP}_\tau = \min \left\{ r_i \mid f_y(\delta(x, M_{r_i})) < \tau \right\}. \tag{6}$$

A low ACP value indicates that minimal masking is sufficient to disrupt the model's prediction, exposing high sensitivity to early perturbations. In contrast, a high ACP suggests that the model maintains confidence across a broader range of input degradation, indicating greater robustness.

## 5 GUIDANCE STRATEGY COMPARISON

The core idea of our framework is to probe model robustness by guiding input degradation with different strategies. Each method produces a saliency map that ranks input regions by estimated importance, and we progressively mask these regions in descending order to obtain a degradation curve $D(r)$. These curves reveal how quickly a model's confidence collapses under different guidance signals. While attribution methods are a natural choice for such guidance, random masking or other heuristics can serve as equally valid baselines. In this view, metrics such as AUBC or ACP do not certify explanatory faithfulness, but rather quantify the behavioral impact of a given guidance strategy on the model. This distinction allows us to compare attribution maps, random masking, and alternative strategies within a unified robustness framework.

### 5.1 IMPLEMENTATION

We implement our framework in PyTorch using a unified forward pass pipeline for all attribution strategies. Each input $x$ is divided into a grid with different sizes. At each perturbation step $t$, the $(t+1)$-th most relevant patch is masked according to a guidance map $g : \mathbb{R}^{H \times W} \to [0, 1]$, and the confidence $f_y(x_t)$ for the ground truth class is recorded. Random masking (A) shuffles patch indices. SHAP (B) computes 1000 Shapley evaluations per image using a custom masker. Grad-CAM (C), SmoothGrad (D, 25 samples), and Integrated Gradients (E, 50 steps) are gradient-based. Activation Maximization (F) optimizes the input over 30 steps. Occlusion Sensitivity (G) perturbs each patch with zeros. Loss Gradient (H) uses the input gradient of the cross-entropy loss.

All saliency maps are normalized to the $[0, 1]$ range, resized to input resolution, and discretized into ranked patch scores. We support multiple perturbation modes: Mean-Distance Perturbation (MDP) which replaces each patch with the maximally distant color in RGB space, zeroing (Black), and white. Gaussian blur was also tested as a perturbation baseline, but ultimately excluded from final experiments, as it failed to cause meaningful confidence degradation. This suggests that blurring may retain enough visual structure for the model to preserve class relevant features, making it ineffective for attribution guided suppression.

### 5.2 LARGE-SCALE EXPERIMENTAL RESULTS

We evaluate eight attribution methods across three datasets: ImageNet-S50 Gao et al. (2022), Oxford Flowers-102 Nilsback & Zisserman (2008), and Oxford-IIIT Pets Parkhi et al. (2012), using five neural network architectures: ResNet32/50 He et al. (2015), VGG16 Simonyan & Zisserman (2015), EfficientNet Tan & Le (2020), DenseNet Huang et al. (2018), and ViT Dosovitskiy et al. (2021). Each architecture is tested under multiple grid sizes and perturbation modes, including MDP, black, and white masking. Gaussian blur was excluded due to negligible effect on prediction behavior. The results are presented in Table 1.

Degradation curves are smoothed using a three parameter sigmoid fit over the 99.999% quantile envelope, following best practices in robust statistics and extreme value modeling Chen & Cosslett (2022); Horbenko et al. (2011); Bensalah (2000); Embrechts et al. (1999). This approach filters local noise while emphasizing sharp confidence drops indicative of model vulnerability. As shown

| Metric | (A) Rand. | (B) SHAP | (C) GC | (D) SC | (E) IG | (F) AM | (G) OS | (H) Loss |
|---|---|---|---|---|---|---|---|---|
| **Imagenet-S50)** | | | | | | | | |
| MDP Perturbation with Patch Size of 14 | | | | | | | | |
| $AUBC^{rel}$ | **0.4655** | 0.6184 | 0.7898 | 0.6989 | 0.8615 | 0.8026 | 0.8952 | 0.8559 |
| $\nabla C^{rel}$ | **0.4658** | 0.5922 | 0.8140 | 0.6926 | 0.8885 | 0.8306 | 0.9150 | 0.8719 |
| $ACP^{rel}_{0.5}$ | **0.4696** | 0.6214 | 0.7252 | 0.7058 | 0.8679 | 0.7996 | 0.9016 | 0.8736 |
| $ACP^{rel}_{0.8}$ | **0.3825** | 0.4829 | 0.6321 | 0.5618 | 0.7665 | 0.6679 | 0.8438 | 0.7903 |
| $AUBC^{abs}$ | **0.6017** | 0.6209 | 0.7524 | 0.6474 | 0.7298 | 0.7303 | 0.8230 | 0.7613 |
| $\nabla C^{abs}$ | **0.6065** | 0.6209 | 0.7649 | 0.6462 | 0.7362 | 0.7395 | 0.8350 | 0.7660 |
| $ACP^{abs}_{0.5}$ | **0.6082** | 0.6258 | 0.7665 | 0.6501 | 0.7373 | 0.7406 | 0.8366 | 0.7693 |
| $ACP^{abs}_{0.8}$ | 0.5348 | **0.5287** | 0.6716 | 0.5701 | 0.6600 | 0.6589 | 0.7748 | 0.6882 |
| Average Metrics across Parameters: MDP-7/14/32, Black-14, White-14 | | | | | | | | |
| $AUBC^{rel}$ | **0.4874** | 0.6484 | 0.8032 | 0.7080 | 0.8594 | 0.8043 | 0.9068 | 0.8651 |
| $\nabla C^{rel}$ | **0.4912** | 0.6413 | 0.7935 | 0.7100 | 0.8916 | 0.8344 | 0.9333 | 0.8906 |
| $ACP^{rel}_{0.5}$ | **0.4951** | 0.6580 | 0.7587 | 0.7042 | 0.8607 | 0.8119 | 0.8933 | 0.8693 |
| $ACP^{rel}_{0.8}$ | **0.4048** | 0.5021 | 0.6455 | 0.5440 | 0.7713 | 0.6724 | 0.8687 | 0.7786 |
| $AUBC^{abs}$ | **0.6380** | 0.6546 | 0.7781 | 0.6729 | 0.7538 | 0.7471 | 0.8376 | 0.7838 |
| $\nabla C^{abs}$ | **0.6455** | 0.6588 | 0.7888 | 0.6766 | 0.7624 | 0.7584 | 0.8519 | 0.7953 |
| $ACP^{abs}_{0.5}$ | **0.6477** | 0.6623 | 0.7907 | 0.6794 | 0.7636 | 0.7601 | 0.8533 | 0.7981 |
| $ACP^{abs}_{0.8}$ | 0.5644 | **0.5609** | 0.7078 | 0.5967 | 0.6988 | 0.6730 | 0.8054 | 0.7286 |
| **Oxford Flowers dataset using ResNet50 MDP with grid sizes 7/14/32, Black-14, and White-14)** | | | | | | | | |
| $AUBC^{rel}$ | **0.5840** | 0.6638 | 0.7016 | 0.6603 | 0.7258 | 0.6794 | 0.9061 | 0.7698 |
| $\nabla C^{rel}$ | **0.5571** | 0.6573 | 0.7028 | 0.6556 | 0.7268 | 0.6780 | 0.9445 | 0.7773 |
| $ACP^{rel}_{0.5}$ | **0.5762** | 0.6697 | 0.7086 | 0.6622 | 0.7301 | 0.6829 | 0.9445 | 0.7781 |
| $ACP^{rel}_{0.8}$ | **0.4238** | 0.4785 | 0.5546 | 0.5190 | 0.6217 | 0.5497 | 0.8634 | 0.6639 |
| $AUBC^{abs}$ | **0.6460** | 0.6494 | 0.7097 | 0.6550 | 0.7288 | 0.6859 | 0.8464 | 0.7557 |
| $\nabla C^{abs}$ | **0.6395** | 0.6400 | 0.7124 | 0.6513 | 0.7303 | 0.6835 | 0.8721 | 0.7618 |
| $ACP^{abs}_{0.5}$ | **0.6461** | 0.6524 | 0.7173 | 0.6571 | 0.7336 | 0.6901 | 0.8721 | 0.7642 |
| $ACP^{abs}_{0.8}$ | 0.5118 | **0.4703** | 0.5790 | 0.5203 | 0.6307 | 0.5467 | 0.7864 | 0.6439 |
| **Oxford-IIIT Pets using ViT-B16 (MDP-14, Black-14 and White-14)** | | | | | | | | |
| $AUBC^{rel}$ | **0.5831** | 0.7161 | 0.9108 | 0.7867 | 0.9469 | 0.9032 | 0.8959 | 0.9052 |
| $\nabla C^{rel}$ | **0.5237** | 0.7246 | 0.9939 | 0.7930 | 0.9729 | 0.9277 | 0.9630 | 0.9133 |
| $ACP^{rel}_{0.5}$ | **0.5679** | 0.7323 | 0.9917 | 0.7974 | 0.9740 | 0.9288 | 0.9597 | 0.9144 |
| $ACP^{rel}_{0.8}$ | **0.3648** | 0.5403 | 0.8350 | 0.6705 | 0.9199 | 0.8339 | 0.8626 | 0.8637 |
| $AUBC^{abs}$ | 0.8081 | **0.7737** | 0.8708 | 0.8080 | 0.8860 | 0.8677 | 0.8517 | 0.8524 |
| $\nabla C^{abs}$ | 0.8129 | **0.7842** | 0.8846 | 0.8162 | 0.8946 | 0.8780 | 0.8990 | 0.8570 |
| $ACP^{abs}_{0.5}$ | 0.8162 | **0.7875** | 0.8879 | 0.8195 | 0.8957 | 0.8813 | 0.8957 | 0.8592 |
| $ACP^{abs}_{0.8}$ | 0.7224 | **0.6484** | 0.8008 | 0.7147 | 0.8372 | 0.7996 | 0.8063 | 0.7886 |

Table 1: Guidance strategy comparison across datasets, models, and perturbations. Columns (A–H) correspond to the guidance strategies described in §5 (A: Random, B: SHAP, C: Grad-CAM, D: SmoothGrad, E: Integrated Gradients, F: Activation Maximization, G: Occlusion Sensitivity, H: Loss Gradient). Superscripts $^{rel}$ and $^{abs}$ denote object-relative and absolute blindness, respectively $ACP_\tau$ uses confidence threshold $\tau$. All metrics are reported on a retained confidence scale, so lower values indicate stronger degradation (i.e., less confidence retained, earlier collapse). **Bold** marks the strongest degradation per row.

in Figure 2, random masking consistently leads to steeper degradation than SHAP, despite lacking any model specific guidance. Figure 3 further illustrates that this trend holds across both object relative and absolute masking settings, underscoring the effectiveness of randomized perturbations.

Across the ImageNet-S50 models, random masking outperforms attribution based methods when perturbations are constrained to annotated object regions. SHAP, Grad-CAM, and Integrated Gradients frequently fail to suppress model confidence effectively in these scenarios. However, when random masking is excluded as a baseline, SHAP consistently yields the best results among attribution based methods. These differences are strongly architecture dependent. Vision Transformers (ViT)

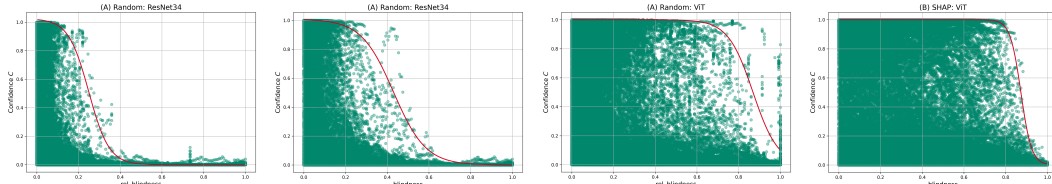

Figure 2: Confidence–blindness trajectories for representative model–guidance pairs with selected ImageNet-S50 model. Each panel shows model confidence $C$ vs. relative and absolute blindness ratio under progressive masking (MDP, patch size 14).

exhibit the highest robustness overall, with smaller gaps between random and attribution guided strategies. This suggests that ViT architectures tend to distribute attention across broader spatial regions. In contrast, convolutional models such as ResNet50 and VGG16 degrade more abruptly, indicating stronger reliance on localized features.

When perturbation is allowed across the entire image (absolute blindness), attribution strategies generally outperform random masking in AUBC$^{abs}$ and ACP$_{0.5}$, although this advantage is not consistent. For instance, on Oxford-IIIT Pets, random masking remains competitive even in the absolute setting. This inconsistency can be attributed to the fact that random masking often removes background regions early in the degradation process, thereby preserving prediction critical regions longer. Attribution based methods, in contrast, typically begin by targeting high importance areas directly, which accelerates confidence decay. These findings caution against over interpreting attribution performance in unconstrained settings.

As illustrated in Figures 2 and 3, saliency maps that appear visually coherent often fail to induce significant prediction collapse. In contrast, random masking—despite being semantically agnostic—frequently causes earlier and more complete degradation. This highlights a fundamental discrepancy: attribution maps may align with human visual intuition, yet misrepresent the model's actual decision dependencies. Object relative masking reveals this discrepancy most clearly, as it confines perturbation to semantically meaningful areas and therefore better exposes alignment between attribution and model behavior.

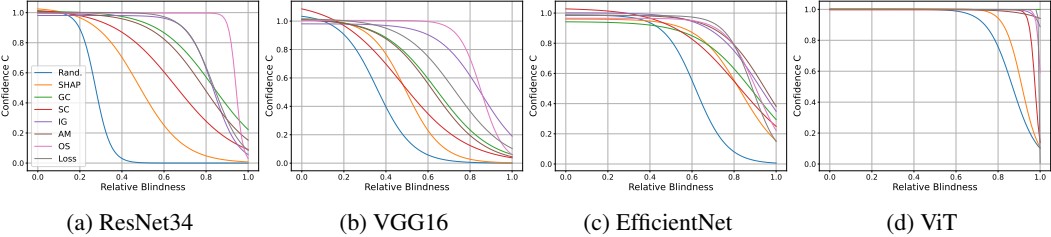

(a) ResNet34     (b) VGG16     (c) EfficientNet     (d) ViT

Figure 3: Confidence degradation trajectories for five representative images from ImageNet-S50 and Oxford Flowers for representative images with MDP perturbation (patch 14). Panels compare ResNet, VGG, EfficientNet, and ViT.

## 5.3 Summary

Random masking consistently produces the strongest degradation across datasets and architectures, particularly under object relative perturbation. This suggests that model predictions are especially sensitive to structural disruptions dispersed across the object area—a property that random masking captures well. Attribution based methods, on the other hand, tend to highlight tightly clustered regions, often centered on prototypical object parts. As shown in Figure 1, this localized masking strategy allows large portions of the object to remain untouched during early perturbation steps, resulting in slower confidence decay. The structural masking bias inherent to most attribution maps

likely explains their underperformance, despite the fact that they appear visually plausible to human observers.

## 6  LIMITATIONS AND DISCUSSION

We test a simple hypothesis: if attribution maps identify regions a model relies on, masking them should maximize degradation. Our finding that a random baseline can induce stronger collapse is not a judgment on explainability—random is not an attribution method—but a diagnostic showing how fragile models can be to indiscriminate evidence removal. The goal of our framework is thus behavioral probing of vulnerability, not ranking methods by human interpretability. We intentionally start with image classification and standard architectures. Extending to multi object detection, segmentation, and video—where structured attribution may be more effective—is important future work. We use widely adopted baselines (Grad-CAM, SHAP, Integrated Gradients) and plan to benchmark newer approaches within the same protocol. Our metrics (AUBC, Sensitivity Slope, ACP) complement deletion/insertion by characterizing how collapse unfolds, not just whether it occurs. Finally, random often outperforms attribution because saliency tends to cluster on prototypical parts, leaving large object regions intact early, whereas random masking erodes evidence more uniformly. Degradation should therefore be read as a behavioral robustness probe—not a proxy for attribution fidelity, with the aim of making models more transparent, predictable, and ultimately safer.

## 7  CONCLUSION

We introduced a degradation based evaluation framework to test the hypothesis that masking the most important regions of an image, as identified by attribution methods, should lead to maximal prediction collapse. By treating attribution maps as guides for structured perturbations, we move beyond visual plausibility and toward measurable behavioral impact. From the resulting degradation curves, we derived three metrics—AUBC, Sensitivity Slope, and Attribution Collapse Point—that quantify how vulnerable model predictions are under progressive masking. Our experiments reveal that random masking often matches or surpasses attribution guided masking. This does not invalidate attribution methods, but shows that degradation is not a direct measure of attribution fidelity. Instead, it highlights that degradation exposes model specific vulnerabilities and interaction effects that saliency alone cannot capture. In this way, degradation serves as a complementary diagnostic tool: it reveals how fragile predictions are to evidence removal and provides insight into where models are most likely to fail. Beyond large scale benchmarking across datasets and architectures, we demonstrated how the framework can uncover class specific weaknesses and support robustness analysis that is both quantitative and reproducible. Looking forward, our approach lays the foundation for integrating degradation metrics into training objectives, ultimately encouraging the design of models that are empirically aware of their own failure modes.

As future work, we envision extending this framework beyond image classification to more complex tasks such as multi object detection, semantic segmentation, and temporal reasoning in video. These settings pose richer challenges for attribution and robustness, where interactions between objects, contexts, and temporal cues play a central role. We see our framework as the starting point for a broader research direction: using maximal degradation as a lens for probing model vulnerability, with the long term goal of making AI systems more transparent, predictable, and ultimately safer in high stakes applications.

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

# A APPENDIX

## APPLICATION EXAMPLE

We demonstrate the practical utility of our degradation framework using the ImageNet-S50 dataset Gao et al. (2022). In this application study, we focus on object relative degradation, using the relative blindness metric $r^{rel}$ to evaluate how much of the object region must be masked before the model's prediction confidence drops significantly. All experiments in this section apply the **random** masking strategy (A) with MDP-14 as a baseline to establish a lower bound reference for model robustness.

### A.1 MODEL COMPARISON

We compare six torchvision models: ResNet18/34/50 He et al. (2015), VGG16, MobileNetV2 Sandler et al. (2019), and ViT-B16—using our robustness metrics: AUBC, sensitivity slope ($\nabla C$), and ACP (Table 4). Results are averaged over all 50 ImageNet-S classes using object relative masking. ViT-B16 shows the highest robustness across all metrics. In contrast, deeper CNNs like ResNet50 and ResNet34 exhibit lower robustness, likely due to reliance on narrow, localized features. Shallower models such as ResNet18 perform better, suggesting that greater depth, while improving accuracy, may increase vulnerability to spatial perturbations. This points to a potential trade off between model capacity and robustness that merits deeper exploration.

### A.2 MODEL EVALUATION

To analyze class specific robustness, we examine degradation curves and metrics for 12 representative ImageNet-S50 classes using ResNet34 (see Figure 4). Some classes, e.g. *Goldfish*, *Tiger Shark* and *Tree Frog*, show sharp confidence drops with minimal masking, indicating brittle reliance on localized cues. Others ,e.g. *Siamese Cat*, *Hamster*, *Ladybug*—degrade more gradually, reflecting more distributed feature use. This variation highlights our framework's ability to reveal category level differences in prediction fragility and identify shortcut prone representations. Such fragilities could guide targeted interventions like occlusion aware augmentation, invariance enforcement, or

| Model | AUBC | $\nabla C$ | $ACP_{0.5}$ | $ACP_{0.8}$ |
|---|---|---|---|---|
| ResNet18 | 0.7697 | 0.7821 | 0.7886 | 0.6549 |
| ResNet34 | 0.6419 | 0.6516 | 0.6516 | 0.6125 |
| ResNet50 | 0.6223 | 0.6255 | 0.6288 | 0.5505 |
| VGG16 | 0.6970 | 0.7005 | 0.7038 | 0.5799 |
| MobileNetV2 | 0.6746 | 0.6712 | 0.6777 | 0.6605 |
| ViT-B16 | 0.8248 | 0.8995 | 0.8929 | 0.7853 |

Table 2: Mean robustness metrics across all 50 classes on ImageNet-S50 using the Random masking (A) baseline. All values are computed using object relative blindness.

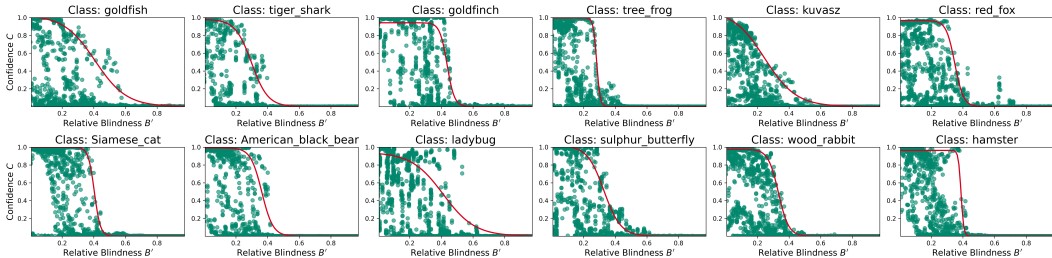

Figure 4: Class-wise degradation curves for the first 12 ImageNet-S50 Gao et al. (2022) classes using ResNet34 He et al. (2015)

saliency based regularization.

| Class: | Goldfish | Tiger Shark | Goldfinch | Tree Fog | Kuvasz | Red Fox |
|---|---|---|---|---|---|---|
| AUBC | 0.4162 | 0.2885 | 0.4056 | 0.2772 | 0.2706 | 0.3326 |
| $\nabla C$ | 0.4050 | 0.2950 | 0.4350 | 0.2850 | 0.2450 | 0.3450 |
| $ACP_{0.5}$ | 0.4150 | 0.3050 | 0.4350 | 0.2850 | 0.2650 | 0.3550 |
| $ACP_{0.8}$ | 0.2850 | 0.2250 | 0.4050 | 0.2750 | 0.1450 | 0.3150 |
| **Class:** | **Siamese cat** | **Black Bear** | **Ladybug** | **Sulphur** | **Wood Rabbit** | **Hamster** |
| AUBC | 0.3982 | 0.3557 | 0.3844 | 0.3301 | 0.3202 | 0.3695 |
| $\nabla C$ | 0.4032 | 0.3650 | 0.4150 | 0.3250 | 0.3317 | 0.3850 |
| $ACP_{0.5}$ | 0.4131 | 0.3650 | 0.4050 | 0.3350 | 0.3317 | 0.3950 |
| $ACP_{0.8}$ | 0.3833 | 0.3150 | 0.2450 | 0.2650 | 0.2921 | 0.3850 |

Table 3: Mean per class robustness metrics for ResNet34 on selected ImageNet-S50 Gao et al. (2022) classes. Values reflect averages over all images within each class, computed using object relative blindness.

## LLM Usage Disclosure

We used a large language model (GPT-5 Thinking via ChatGPT) strictly for grammar and style polishing of text written by the authors. The model did not generate new scientific content, claims, analysis, figures, tables or references. All technical content and conclusions are the authors' own, and the authors verified every change. Any errors remain our responsibility.

## Experimental Results - Guidance Comparison

| Model | Metric | (A) Random | (B) SHAP | (C) GC | (D) SC | (E) IG | (F) AM | (G) OS | (H) Loss |
|---|---|---|---|---|---|---|---|---|---|
| ResNet34 | $\text{AUBC}^{rel}$ | **0.2504** | 0.4725 | 0.5403 | 0.5908 | 0.7769 | 0.7371 | 0.7944 | 0.8121 |
| | $\nabla C^{rel}$ | **0.2500** | 0.4553 | 0.5182 | 0.5778 | 0.7963 | 0.7467 | 0.8096 | 0.8361 |
| | $\text{ACP}_{0.5}^{rel}$ | **0.2533** | 0.4652 | 0.5315 | 0.5877 | 0.7963 | 0.7500 | 0.8162 | 0.8361 |
| | $\text{ACP}_{0.8}^{rel}$ | **0.1871** | 0.3328 | 0.3858 | 0.4387 | 0.7367 | 0.6076 | 0.6507 | 0.7301 |
| | $\text{AUBC}^{abs}$ | 0.4293 | **0.4142** | 0.5847 | 0.5443 | 0.6267 | 0.6242 | 0.6824 | 0.6977 |
| | $\nabla C^{abs}$ | 0.4288 | **0.4123** | 0.5844 | 0.5348 | 0.6308 | 0.6242 | 0.6805 | 0.7003 |
| | $\text{ACP}_{0.5}^{abs}$ | 0.4321 | **0.4156** | 0.5877 | 0.5414 | 0.6308 | 0.6275 | 0.6838 | 0.7036 |
| | $\text{ACP}_{0.8}^{abs}$ | **0.3295** | 0.3493 | 0.5017 | 0.4354 | 0.5215 | 0.5182 | 0.5480 | 0.5911 |
| ResNet50 | $\text{AUBC}^{rel}$ | **0.2713** | 0.4917 | 0.8040 | 0.6532 | 0.8099 | 0.7631 | 0.9340 | 0.8283 |
| | $\nabla C^{rel}$ | **0.2732** | 0.4785 | 0.8361 | 0.6540 | 0.8328 | 0.7897 | 0.9420 | 0.8394 |
| | $\text{ACP}_{0.5}^{rel}$ | **0.2765** | 0.4884 | 0.8394 | 0.6639 | 0.8328 | 0.7897 | 0.9453 | 0.8394 |
| | $\text{ACP}_{0.8}^{rel}$ | **0.2268** | 0.3527 | 0.6606 | 0.4685 | 0.7467 | 0.6209 | 0.9222 | 0.7467 |
| | $\text{AUBC}^{abs}$ | **0.4549** | 0.5127 | 0.6861 | 0.5573 | 0.6495 | 0.6754 | 0.8218 | 0.7041 |
| | $\nabla C^{abs}$ | **0.4619** | 0.4983 | 0.6904 | 0.5414 | 0.6473 | 0.6871 | 0.8294 | 0.7069 |
| | $\text{ACP}_{0.5}^{abs}$ | **0.4619** | 0.5083 | 0.6904 | 0.5513 | 0.6507 | 0.6871 | 0.8294 | 0.7103 |
| | $\text{ACP}_{0.8}^{abs}$ | 0.4023 | **0.3791** | 0.6142 | 0.4156 | 0.5447 | 0.6043 | 0.8096 | 0.6175 |
| VGG16 | $\text{AUBC}^{rel}$ | **0.3984** | 0.4992 | 0.6425 | 0.5341 | 0.8095 | 0.6237 | 0.8493 | 0.7225 |
| | $\nabla C^{rel}$ | **0.3957** | 0.4950 | 0.6440 | 0.4719 | 0.8526 | 0.6242 | 0.8526 | 0.7268 |
| | $\text{ACP}_{0.5}^{rel}$ | **0.3990** | 0.4983 | 0.6507 | 0.5116 | 0.8526 | 0.6308 | 0.8559 | 0.7334 |
| | $\text{ACP}_{0.8}^{rel}$ | **0.3394** | 0.3858 | 0.4785 | 0.3394 | 0.7003 | 0.4652 | 0.7831 | 0.5679 |
| | $\text{AUBC}^{abs}$ | **0.4088** | 0.5597 | 0.6154 | 0.4744 | 0.6697 | 0.6226 | 0.7871 | 0.6694 |
| | $\nabla C^{abs}$ | **0.4089** | 0.5613 | 0.6109 | 0.4685 | 0.6771 | 0.6275 | 0.7996 | 0.6639 |
| | $\text{ACP}_{0.5}^{abs}$ | **0.4123** | 0.5646 | 0.6175 | 0.4752 | 0.6771 | 0.6308 | 0.7996 | 0.6705 |
| | $\text{ACP}_{0.8}^{abs}$ | **0.3593** | 0.4586 | 0.4884 | 0.3924 | 0.5977 | 0.5182 | 0.7566 | 0.5248 |
| EfficientNet | $\text{AUBC}^{rel}$ | 0.6051 | **0.5649** | 0.7903 | 0.7962 | 0.8720 | 0.8785 | 0.8535 | 0.8592 |
| | $\nabla C^{rel}$ | 0.6175 | **0.3791** | 0.8957 | 0.8195 | 0.9255 | 0.9486 | 0.9089 | 0.8890 |
| | $\text{ACP}_{0.5}^{rel}$ | 0.6175 | **0.5348** | 0.8791 | 0.8328 | 0.9288 | 0.9486 | 0.9056 | 0.8890 |
| | $\text{ACP}_{0.8}^{rel}$ | 0.5116 | **0.2964** | 0.6672 | 0.6275 | 0.7665 | 0.7930 | 0.7930 | 0.7963 |
| | $\text{AUBC}^{abs}$ | 0.7087 | **0.6308** | 0.8307 | 0.8720 | 0.7516 | 0.8189 | 0.8045 | 0.8092 |
| | $\nabla C^{abs}$ | 0.7235 | **0.6109** | 0.8957 | 0.9255 | 0.7632 | 0.8427 | 0.8427 | 0.8228 |
| | $\text{ACP}_{0.5}^{abs}$ | 0.7235 | **0.6242** | 0.8924 | 0.9288 | 0.7632 | 0.8427 | 0.8427 | 0.8228 |
| | $\text{ACP}_{0.8}^{abs}$ | 0.6341 | **0.4652** | 0.7103 | 0.7665 | 0.6838 | 0.7765 | 0.7500 | 0.7665 |
| DenseNet | $\text{AUBC}^{rel}$ | **0.4675** | 0.5649 | 0.9723 | 0.6537 | 0.9138 | 0.8288 | 0.9543 | 0.9263 |
| | $\nabla C^{rel}$ | 0.4752 | **0.3791** | 0.9950 | 0.6573 | 0.9288 | 0.8791 | 0.9818 | 0.9453 |
| | $\text{ACP}_{0.5}^{rel}$ | **0.4752** | 0.5348 | – | 0.6606 | 0.9288 | 0.8791 | 0.9851 | 0.9486 |
| | $\text{ACP}_{0.8}^{rel}$ | 0.4487 | **0.2964** | 0.9685 | 0.5381 | 0.8824 | 0.8526 | 0.9222 | 0.9056 |
| | $\text{AUBC}^{abs}$ | **0.4876** | 0.6308 | 0.8457 | 0.6556 | 0.7476 | 0.7242 | 0.8775 | 0.7909 |
| | $\nabla C^{abs}$ | **0.4950** | 0.6109 | 0.8493 | 0.6606 | 0.7566 | 0.7334 | 0.8857 | 0.7996 |
| | $\text{ACP}_{0.5}^{abs}$ | **0.4950** | 0.6242 | 0.8526 | 0.6606 | 0.7566 | 0.7334 | 0.8890 | 0.8030 |
| | $\text{ACP}_{0.8}^{abs}$ | **0.4553** | 0.4652 | 0.7698 | 0.5811 | 0.7069 | 0.6407 | 0.8228 | 0.7334 |
| ViT | $\text{AUBC}^{rel}$ | **0.8570** | 0.9059 | 0.9897 | 0.9655 | 0.9870 | 0.9844 | 0.9856 | 0.9872 |
| | $\nabla C^{rel}$ | **0.8692** | 0.9122 | 0.9950 | 0.9751 | 0.9950 | 0.9950 | 0.9950 | 0.9950 |
| | $\text{ACP}_{0.5}^{rel}$ | **0.8692** | 0.9155 | – | 0.9784 | – | – | – | 0.9950 |
| | $\text{ACP}_{0.8}^{rel}$ | **0.7864** | 0.8592 | – | 0.9586 | – | – | 0.9917 | 0.9950 |
| | $\text{AUBC}^{abs}$ | 0.8826 | **0.8681** | 0.9515 | 0.9071 | 0.9338 | 0.9164 | 0.9649 | 0.8966 |
| | $\nabla C^{abs}$ | 0.8890 | **0.8725** | 0.9586 | 0.9155 | 0.9420 | 0.9222 | 0.9718 | 0.9023 |
| | $\text{ACP}_{0.5}^{abs}$ | 0.8890 | **0.8758** | 0.9586 | 0.9155 | 0.9453 | 0.9222 | 0.9751 | 0.9056 |
| | $\text{ACP}_{0.8}^{abs}$ | 0.8626 | **0.8394** | 0.9453 | 0.8990 | 0.9056 | 0.8957 | 0.9619 | 0.8957 |

Table 4: Quantitative robustness evaluation across models, datasets, and attribution methods with **MDP Perturbation** and **cell size of 14**. Dashes (**–**) indicate that the confidence threshold was not reached

| Model | Metric | (A) Random | (B) SHAP | (C) GC | (D) SC | (E) IG | (F) AM | (G) OS | (H) Loss |
|---|---|---|---|---|---|---|---|---|---|
| ResNet34 | $\mathrm{AUBC}^{rel}$ | 0.5582 | **0.5203** | 0.5525 | 0.6770 | 0.8494 | 0.7898 | 0.9228 | 0.8844 |
| | $\nabla C^{rel}$ | 0.5348 | **0.5149** | 0.5447 | 0.6871 | 0.8924 | 0.8394 | 0.9884 | 0.9155 |
| | $\mathrm{ACP}^{rel}_{0.5}$ | 0.5480 | **0.5215** | 0.5513 | 0.6871 | 0.8890 | 0.8328 | 0.9884 | 0.9155 |
| | $\mathrm{ACP}^{rel}_{0.8}$ | 0.4056 | **0.3891** | 0.4089 | 0.5381 | 0.8096 | 0.6904 | 0.8824 | 0.8427 |
| | $\mathrm{AUBC}^{abs}$ | 0.6010 | **0.5400** | 0.6315 | 0.7155 | 0.7311 | 0.7383 | 0.8378 | 0.8110 |
| | $\nabla C^{abs}$ | 0.5987 | **0.5338** | 0.6279 | 0.7220 | 0.7414 | 0.7447 | 0.8517 | 0.8225 |
| | $\mathrm{ACP}^{abs}_{0.5}$ | 0.6019 | **0.5403** | 0.6311 | 0.7220 | 0.7414 | 0.7479 | 0.8517 | 0.8258 |
| | $\mathrm{ACP}^{abs}_{0.8}$ | 0.5046 | **0.4365** | 0.5273 | 0.6376 | 0.6473 | 0.6636 | 0.8063 | 0.7479 |
| ResNet50 | $\mathrm{AUBC}^{rel}$ | 0.6276 | **0.5724** | 0.7256 | 0.6751 | 0.8681 | 0.7786 | 0.9869 | 0.8855 |
| | $\nabla C^{rel}$ | 0.6275 | **0.5149** | 0.7268 | 0.6672 | 0.9155 | 0.7996 | 0.9950 | 0.9023 |
| | $\mathrm{ACP}^{rel}_{0.5}$ | 0.6308 | **0.5546** | 0.7599 | 0.6838 | 0.9122 | 0.8030 | – | 0.9023 |
| | $\mathrm{ACP}^{rel}_{0.8}$ | 0.4950 | **0.3626** | 0.4983 | 0.4785 | 0.7963 | 0.6606 | 0.9950 | 0.8195 |
| | $\mathrm{AUBC}^{abs}$ | 0.7103 | **0.6165** | 0.7064 | 0.6852 | 0.7824 | 0.7619 | 0.9196 | 0.7888 |
| | $\nabla C^{abs}$ | 0.7187 | **0.6246** | 0.7155 | 0.6895 | 0.7933 | 0.7641 | 0.9263 | 0.7966 |
| | $\mathrm{ACP}^{abs}_{0.5}$ | 0.7187 | **0.6246** | 0.7155 | 0.6928 | 0.7966 | 0.7674 | 0.9296 | 0.7998 |
| | $\mathrm{ACP}^{abs}_{0.8}$ | 0.6571 | **0.5241** | 0.6571 | 0.6084 | 0.7155 | 0.6928 | 0.9166 | 0.7447 |
| VGG16 | $\mathrm{AUBC}^{rel}$ | 0.7821 | **0.6276** | 0.6337 | 0.6274 | 0.8146 | 0.7280 | 0.8841 | 0.7781 |
| | $\nabla C^{rel}$ | 0.7831 | 0.6573 | **0.6142** | 0.6076 | 0.8824 | 0.7467 | 0.9122 | 0.7996 |
| | $\mathrm{ACP}^{rel}_{0.5}$ | 0.7864 | 0.6507 | 0.6374 | **0.6242** | 0.8957 | 0.7500 | 0.9155 | 0.8063 |
| | $\mathrm{ACP}^{rel}_{0.8}$ | 0.6838 | 0.4884 | **0.4222** | 0.4520 | 0.6374 | 0.5546 | 0.8096 | 0.6209 |
| | $\mathrm{AUBC}^{abs}$ | 0.7249 | 0.6892 | 0.6766 | **0.6631** | 0.7110 | 0.6959 | 0.8326 | 0.7595 |
| | $\nabla C^{abs}$ | 0.7220 | 0.7122 | **0.6830** | 0.6668 | 0.7122 | 0.7025 | 0.8452 | 0.7674 |
| | $\mathrm{ACP}^{abs}_{0.5}$ | 0.7252 | 0.7122 | 0.6863 | **0.6701** | 0.7155 | 0.7025 | 0.8485 | 0.7706 |
| | $\mathrm{ACP}^{abs}_{0.8}$ | 0.6538 | 0.6311 | 0.5954 | **0.5597** | 0.6571 | 0.6279 | 0.7868 | 0.6765 |
| EfficientNet | $\mathrm{AUBC}^{rel}$ | 0.7308 | 0.7680 | **0.7248** | 0.8284 | 0.8682 | 0.8637 | 0.9436 | 0.8770 |
| | $\nabla C^{rel}$ | **0.7500** | 0.8328 | 0.7864 | 0.9023 | 0.9950 | 0.9718 | 0.9950 | 0.9950 |
| | $\mathrm{ACP}^{rel}_{0.5}$ | **0.7500** | 0.8361 | 0.7765 | 0.9023 | – | 0.9718 | – | – |
| | $\mathrm{ACP}^{rel}_{0.8}$ | 0.6109 | 0.5811 | **0.5613** | 0.7003 | 0.7334 | 0.7500 | 0.9553 | 0.7665 |
| | $\mathrm{AUBC}^{abs}$ | **0.7692** | 0.7839 | 0.7998 | 0.7874 | 0.8068 | 0.8164 | 0.9054 | 0.8303 |
| | $\nabla C^{abs}$ | **0.7804** | 0.8095 | 0.8258 | 0.8031 | 0.8258 | 0.8323 | 0.9296 | 0.8550 |
| | $\mathrm{ACP}^{abs}_{0.5}$ | **0.7836** | 0.8095 | 0.8258 | 0.8031 | 0.8258 | 0.8355 | 0.9296 | 0.8550 |
| | $\mathrm{ACP}^{abs}_{0.8}$ | **0.6798** | **0.6798** | 0.7155 | 0.7317 | 0.7544 | 0.7317 | 0.9004 | 0.7901 |
| DenseNet | $\mathrm{AUBC}^{rel}$ | 0.8544 | **0.5954** | 0.9528 | 0.7276 | 0.9477 | 0.8397 | 0.9837 | 0.9370 |
| | $\nabla C^{rel}$ | 0.8592 | **0.5579** | 0.9950 | 0.7467 | 0.9950 | 0.9288 | 0.9950 | 0.9619 |
| | $\mathrm{ACP}^{rel}_{0.5}$ | 0.8626 | **0.5877** | – | 0.7500 | – | 0.9354 | – | 0.9652 |
| | $\mathrm{ACP}^{rel}_{0.8}$ | 0.7864 | **0.3924** | 0.9321 | 0.5646 | 0.9586 | 0.6871 | – | 0.8957 |
| | $\mathrm{AUBC}^{abs}$ | 0.8327 | **0.6609** | 0.8493 | 0.7897 | 0.7987 | 0.7875 | 0.9243 | 0.8477 |
| | $\nabla C^{abs}$ | 0.8387 | **0.6603** | 0.8550 | 0.7998 | 0.8063 | 0.7901 | 0.9328 | 0.8582 |
| | $\mathrm{ACP}^{abs}_{0.5}$ | 0.8420 | **0.6636** | 0.8550 | 0.7998 | 0.8063 | 0.7901 | 0.9328 | 0.8582 |
| | $\mathrm{ACP}^{abs}_{0.8}$ | 0.7771 | **0.5597** | 0.7966 | 0.7414 | 0.7544 | 0.7220 | 0.9134 | 0.8128 |
| ViT | $\mathrm{AUBC}^{rel}$ | 0.9260 | **0.9189** | 0.9896 | 0.9856 | 0.9876 | 0.9832 | 0.9859 | 0.9891 |
| | $\nabla C^{rel}$ | 0.9751 | **0.9486** | – | 0.9950 | 0.9950 | 0.9950 | 0.9950 | 0.9950 |
| | $\mathrm{ACP}^{rel}_{0.5}$ | 0.9784 | **0.9486** | – | – | – | – | – | – |
| | $\mathrm{ACP}^{rel}_{0.8}$ | 0.8592 | **0.8659** | – | – | – | – | – | – |
| | $\mathrm{AUBC}^{abs}$ | **0.8873** | 0.9093 | 0.9624 | 0.9318 | 0.9447 | 0.9416 | 0.9620 | 0.9419 |
| | $\nabla C^{abs}$ | **0.8939** | 0.9198 | 0.9718 | 0.9393 | 0.9555 | 0.9555 | 0.9718 | 0.9490 |
| | $\mathrm{ACP}^{abs}_{0.5}$ | **0.8971** | 0.9231 | 0.9718 | 0.9393 | 0.9588 | 0.9555 | 0.9718 | 0.9490 |
| | $\mathrm{ACP}^{abs}_{0.8}$ | **0.8615** | 0.8874 | 0.9653 | 0.9263 | 0.9393 | 0.9393 | 0.9620 | 0.9393 |

Table 5: ImageNet-S: Quantitative robustness evaluation across models, datasets, and attribution methods with **MDP Perturbation** and **cell size of 7**. Dashes (**–**) indicate that the confidence threshold was not reached

| Model | Metric | (A) Random | (B) SHAP | (C) GC | (D) SC | (E) IG | (F) AM | (G) OS | (H) Loss |
|---|---|---|---|---|---|---|---|---|---|
| ResNet34 | $AUBC^{rel}$ | **0.0803** | 0.4410 | 0.5937 | 0.4669 | 0.7724 | 0.6630 | 0.6442 | 0.7513 |
| | $\nabla C^{rel}$ | **0.0845** | 0.4421 | 0.5944 | 0.4156 | 0.7996 | 0.7202 | 0.6805 | 0.7765 |
| | $ACP_{0.5}^{rel}$ | **0.0878** | 0.4454 | 0.5944 | 0.4487 | 0.7963 | 0.7069 | 0.6805 | 0.8030 |
| | $ACP_{0.8}^{rel}$ | **0.0613** | 0.3295 | 0.5017 | 0.2765 | 0.7169 | 0.5215 | 0.6705 | 0.5315 |
| | $AUBC^{abs}$ | **0.3368** | 0.3962 | 0.7101 | 0.4097 | 0.6305 | 0.5552 | 0.5831 | 0.6140 |
| | $\nabla C^{abs}$ | **0.3626** | 0.3891 | 0.7036 | 0.3825 | 0.6407 | 0.5646 | 0.5877 | 0.6407 |
| | $ACP_{0.5}^{abs}$ | **0.3593** | 0.3957 | 0.7136 | 0.3990 | 0.6407 | 0.5646 | 0.5877 | 0.6407 |
| | $ACP_{0.8}^{abs}$ | **0.2831** | 0.2964 | 0.5480 | 0.2798 | 0.5281 | 0.4719 | 0.5315 | 0.5712 |
| ResNet50 | $AUBC^{rel}$ | **0.1252** | 0.4859 | 0.7692 | 0.4177 | 0.7568 | 0.6702 | 0.7700 | 0.7793 |
| | $\nabla C^{rel}$ | **0.1143** | 0.4454 | 0.7698 | 0.4023 | 0.7831 | 0.7169 | 0.7798 | 0.7930 |
| | $ACP_{0.5}^{rel}$ | **0.1242** | 0.4719 | 0.7732 | 0.4123 | 0.7831 | 0.7069 | 0.7831 | 0.7963 |
| | $ACP_{0.8}^{rel}$ | **0.0778** | 0.2997 | 0.6540 | 0.2931 | 0.6771 | 0.5447 | 0.7632 | 0.6507 |
| | $AUBC^{abs}$ | **0.3948** | 0.4662 | 0.6708 | 0.3362 | 0.5757 | 0.5130 | 0.6687 | 0.5643 |
| | $\nabla C^{abs}$ | **0.4089** | 0.4619 | 0.6771 | 0.3361 | 0.5712 | 0.5215 | 0.6771 | 0.5679 |
| | $ACP_{0.5}^{abs}$ | **0.4056** | 0.4685 | 0.6805 | 0.3394 | 0.5745 | 0.5215 | 0.6805 | 0.5712 |
| | $ACP_{0.8}^{abs}$ | **0.3063** | 0.3758 | 0.6076 | 0.2666 | 0.4818 | 0.3891 | 0.6573 | 0.5116 |
| VGG16 | $AUBC^{rel}$ | **0.1717** | 0.4566 | 0.7571 | 0.4180 | 0.6564 | 0.5369 | 0.7371 | 0.6667 |
| | $\nabla C^{rel}$ | **0.1772** | 0.4387 | 0.8030 | 0.2599 | 0.6606 | 0.4685 | 0.7434 | 0.6275 |
| | $ACP_{0.5}^{rel}$ | **0.1772** | 0.4487 | 0.7963 | 0.3791 | 0.6639 | 0.5215 | 0.7434 | 0.6838 |
| | $ACP_{0.8}^{rel}$ | **0.1374** | 0.3295 | 0.6705 | 0.2004 | 0.5712 | 0.2997 | 0.7434 | 0.4123 |
| | $AUBC^{abs}$ | **0.1746** | 0.5083 | 0.6367 | 0.3456 | 0.5564 | 0.4995 | 0.7496 | 0.5741 |
| | $\nabla C^{abs}$ | **0.1805** | 0.5083 | 0.6407 | 0.3394 | 0.5712 | 0.4785 | 0.7533 | 0.5281 |
| | $ACP_{0.5}^{abs}$ | **0.1805** | 0.5116 | 0.6407 | 0.3460 | 0.5712 | 0.4917 | 0.7566 | 0.5646 |
| | $ACP_{0.8}^{abs}$ | **0.1441** | 0.4089 | 0.5745 | 0.2566 | 0.5480 | 0.3493 | 0.7367 | 0.3527 |
| EfficientNet | $AUBC^{rel}$ | **0.3095** | 0.7573 | 0.7619 | 0.7828 | 0.8488 | 0.8337 | 0.8747 | 0.7959 |
| | $\nabla C^{rel}$ | **0.3262** | 0.7996 | 0.8162 | 0.8261 | 0.9056 | 0.9089 | 0.9188 | 0.8559 |
| | $ACP_{0.5}^{rel}$ | **0.3262** | 0.7996 | 0.8096 | 0.8592 | 0.9056 | 0.9023 | 0.9188 | 0.8526 |
| | $ACP_{0.8}^{rel}$ | **0.2964** | 0.6010 | 0.6341 | 0.5679 | 0.7434 | 0.7500 | 0.8592 | 0.7930 |
| | $AUBC^{abs}$ | **0.5947** | 0.6497 | 0.7645 | 0.6799 | 0.6535 | 0.6835 | 0.8099 | 0.6766 |
| | $\nabla C^{abs}$ | **0.6242** | 0.6606 | 0.7798 | 0.6904 | 0.6705 | 0.7136 | 0.8427 | 0.6937 |
| | $ACP_{0.5}^{abs}$ | **0.6209** | 0.6606 | 0.7864 | 0.6937 | 0.6705 | 0.7136 | 0.8427 | 0.6937 |
| | $ACP_{0.8}^{abs}$ | 0.5381 | **0.5315** | 0.6175 | 0.5281 | 0.5977 | 0.6871 | 0.7467 | 0.5977 |
| DenseNet | $AUBC^{rel}$ | **0.2432** | 0.5496 | 0.9684 | 0.3977 | 0.7489 | 0.7232 | 0.9655 | 0.6723 |
| | $\nabla C^{rel}$ | **0.2467** | 0.5215 | 0.9818 | 0.4023 | 0.7533 | 0.7434 | 0.9950 | 0.6970 |
| | $ACP_{0.5}^{rel}$ | **0.2500** | 0.5414 | 0.9818 | 0.4056 | 0.7533 | 0.7434 | – | 0.7003 |
| | $ACP_{0.8}^{rel}$ | **0.2335** | 0.3626 | 0.9619 | 0.3394 | 0.7003 | 0.6937 | 0.9718 | 0.6838 |
| | $AUBC^{abs}$ | **0.2422** | 0.5731 | 0.8514 | 0.5583 | 0.7555 | 0.7098 | 0.9281 | 0.5036 |
| | $\nabla C^{abs}$ | **0.2467** | 0.5844 | 0.8559 | 0.5712 | 0.7632 | 0.7169 | 0.9486 | 0.5215 |
| | $ACP_{0.5}^{abs}$ | **0.2500** | 0.5844 | 0.8592 | 0.5712 | 0.7665 | 0.7169 | 0.9486 | 0.5215 |
| | $ACP_{0.8}^{abs}$ | **0.2302** | 0.5050 | 0.7732 | 0.5348 | 0.7434 | 0.6573 | 0.9321 | 0.5017 |
| ViT | $AUBC^{rel}$ | **0.1529** | 0.7773 | 0.9890 | 0.7634 | 0.9236 | 0.7474 | 0.8219 | 0.8885 |
| | $\nabla C^{rel}$ | **0.1507** | 0.7698 | 0.9950 | 0.7864 | 0.9950 | 0.7599 | 0.8361 | 0.8990 |
| | $ACP_{0.5}^{rel}$ | **0.1573** | 0.7765 | – | 0.7864 | – | 0.7599 | 0.8361 | 0.9023 |
| | $ACP_{0.8}^{rel}$ | **0.1110** | 0.6838 | – | 0.7069 | 0.8592 | 0.7136 | 0.8162 | 0.8361 |
| | $AUBC^{abs}$ | **0.5475** | 0.6794 | 0.8940 | 0.4932 | 0.6437 | 0.6185 | 0.6013 | 0.6887 |
| | $\nabla C^{abs}$ | **0.5745** | 0.6738 | 0.9056 | 0.4983 | 0.6507 | 0.6308 | 0.6142 | 0.6937 |
| | $ACP_{0.5}^{abs}$ | **0.5745** | 0.6771 | 0.9056 | 0.4983 | 0.6507 | 0.6308 | 0.6142 | 0.6970 |
| | $ACP_{0.8}^{abs}$ | **0.5348** | 0.6043 | 0.8824 | 0.4454 | 0.6242 | 0.5977 | 0.5844 | 0.6573 |

Table 6: ImageNet-S: Quantitative robustness evaluation across models, datasets, and attribution methods with **MDP Perturbation** and **cell size of 32**. Dashes (**–**) indicate that the confidence threshold was not reached

| Model | Metric | (A) Random | (B) SHAP | (C) GC | (D) SC | (E) IG | (F) AM | (G) OS | (H) Loss |
|---|---|---|---|---|---|---|---|---|---|
| ResNet34 | $\mathrm{AUBC}^{rel}$ | **0.4081** | 0.6372 | 0.6985 | 0.7293 | 0.8014 | 0.8052 | 0.8935 | 0.9254 |
| | $\nabla C^{rel}$ | **0.4156** | 0.6341 | 0.7003 | 0.7367 | 0.8162 | 0.8294 | 0.9321 | 0.9685 |
| | $\mathrm{ACP}^{rel}_{0.5}$ | **0.4156** | 0.6407 | 0.7036 | 0.7434 | 0.8162 | 0.8361 | 0.9321 | 0.9685 |
| | $\mathrm{ACP}^{rel}_{0.8}$ | **0.3229** | 0.4851 | 0.5877 | 0.5579 | 0.7533 | 0.6639 | 0.9056 | 0.9023 |
| | $\mathrm{AUBC}^{abs}$ | **0.5841** | 0.5986 | 0.7675 | 0.7306 | 0.7579 | 0.7806 | 0.7829 | 0.8742 |
| | $\nabla C^{abs}$ | **0.5844** | 0.5944 | 0.7765 | 0.7367 | 0.7665 | 0.7930 | 0.7996 | 0.9122 |
| | $\mathrm{ACP}^{abs}_{0.5}$ | **0.5877** | 0.5977 | 0.7765 | 0.7401 | 0.7698 | 0.7996 | 0.7996 | 0.9122 |
| | $\mathrm{ACP}^{abs}_{0.8}$ | **0.4851** | 0.5050 | 0.7202 | 0.6308 | 0.6771 | 0.6473 | 0.6970 | 0.8758 |
| ResNet50 | $\mathrm{AUBC}^{rel}$ | **0.3692** | 0.5307 | 0.8164 | 0.6846 | 0.8362 | 0.8641 | 0.9685 | 0.9428 |
| | $\nabla C^{rel}$ | **0.3791** | 0.5248 | 0.8592 | 0.6838 | 0.8526 | 0.9056 | 0.9818 | 0.9950 |
| | $\mathrm{ACP}^{rel}_{0.5}$ | **0.3791** | 0.5315 | 0.8626 | 0.6904 | 0.8526 | 0.9089 | 0.9851 | – |
| | $\mathrm{ACP}^{rel}_{0.8}$ | **0.3162** | 0.4354 | 0.6705 | 0.5348 | 0.7963 | 0.7732 | 0.9586 | 0.9122 |
| | $\mathrm{AUBC}^{abs}$ | **0.6060** | 0.6222 | 0.7646 | 0.7218 | 0.7362 | 0.8297 | 0.8790 | 0.8904 |
| | $\nabla C^{abs}$ | **0.6109** | 0.6242 | 0.7698 | 0.7401 | 0.7467 | 0.8626 | 0.8857 | 0.9288 |
| | $\mathrm{ACP}^{abs}_{0.5}$ | **0.6142** | 0.6275 | 0.7732 | 0.7401 | 0.7467 | 0.8626 | 0.8890 | 0.9288 |
| | $\mathrm{ACP}^{abs}_{0.8}$ | **0.5083** | 0.4950 | 0.7003 | 0.6010 | 0.6507 | 0.7897 | 0.8526 | 0.8692 |
| VGG16 | $\mathrm{AUBC}^{rel}$ | **0.5433** | 0.5649 | 0.6236 | 0.6111 | 0.8029 | 0.6948 | 0.9545 | 0.7993 |
| | $\nabla C^{rel}$ | **0.5381** | 0.5447 | 0.6142 | 0.5911 | 0.8460 | 0.7103 | 0.9950 | 0.8361 |
| | $\mathrm{ACP}^{rel}_{0.5}$ | **0.5447** | 0.5579 | 0.6242 | 0.6076 | 0.8460 | 0.7103 | – | 0.8427 |
| | $\mathrm{ACP}^{rel}_{0.8}$ | 0.4255 | **0.3990** | 0.4454 | 0.4354 | 0.6705 | 0.5348 | 0.9354 | 0.6440 |
| | $\mathrm{AUBC}^{abs}$ | **0.5352** | 0.6089 | 0.6195 | 0.5783 | 0.7017 | 0.7268 | 0.8780 | 0.7723 |
| | $\nabla C^{abs}$ | **0.5315** | 0.6076 | 0.6142 | 0.5745 | 0.7003 | 0.7533 | 0.8890 | 0.7963 |
| | $\mathrm{ACP}^{abs}_{0.5}$ | **0.5348** | 0.6109 | 0.6175 | 0.5778 | 0.7036 | 0.7500 | 0.8924 | 0.7963 |
| | $\mathrm{ACP}^{abs}_{0.8}$ | **0.4354** | 0.4851 | 0.4884 | 0.4652 | 0.6109 | 0.6109 | 0.8261 | 0.6473 |
| EfficientNet | $\mathrm{AUBC}^{rel}$ | **0.7056** | 0.7658 | 0.7400 | 0.8045 | 0.8691 | 0.8662 | 0.8839 | 0.8487 |
| | $\nabla C^{rel}$ | **0.7235** | 0.8063 | 0.7963 | 0.8228 | 0.9056 | 0.9188 | 0.9486 | 0.8692 |
| | $\mathrm{ACP}^{rel}_{0.5}$ | **0.7235** | 0.8063 | 0.7897 | 0.8294 | 0.9056 | 0.9155 | 0.9453 | 0.8692 |
| | $\mathrm{ACP}^{rel}_{0.8}$ | **0.5977** | 0.6242 | 0.6076 | 0.6738 | 0.7897 | 0.7930 | 0.8692 | 0.7732 |
| | $\mathrm{AUBC}^{abs}$ | 0.8030 | **0.7304** | 0.8104 | 0.7387 | 0.7933 | 0.8331 | 0.8152 | 0.8353 |
| | $\nabla C^{abs}$ | 0.8294 | **0.7533** | 0.8328 | 0.7500 | 0.8063 | 0.8526 | 0.8526 | 0.8526 |
| | $\mathrm{ACP}^{abs}_{0.5}$ | 0.8294 | **0.7533** | 0.8328 | 0.7500 | 0.8096 | 0.8526 | 0.8526 | 0.8526 |
| | $\mathrm{ACP}^{abs}_{0.8}$ | 0.7235 | **0.6175** | 0.7533 | 0.6937 | 0.7367 | 0.7665 | 0.7765 | 0.7996 |
| DenseNet | $\mathrm{AUBC}^{rel}$ | **0.5195** | 0.6156 | 0.9792 | 0.6858 | 0.9176 | 0.8540 | 0.9720 | 0.9421 |
| | $\nabla C^{rel}$ | 0.5083 | **0.4884** | 0.9950 | 0.7103 | 0.9288 | 0.8791 | 0.9950 | 0.9851 |
| | $\mathrm{ACP}^{rel}_{0.5}$ | 0.5149 | 0.6010 | – | 0.7069 | 0.9288 | 0.8791 | – | 0.9851 |
| | $\mathrm{ACP}^{rel}_{0.8}$ | 0.4222 | **0.3493** | 0.9818 | 0.5745 | 0.8890 | 0.7632 | – | 0.8990 |
| | $\mathrm{AUBC}^{abs}$ | **0.5119** | 0.6911 | 0.8961 | 0.7255 | 0.8227 | 0.8020 | 0.9009 | 0.8548 |
| | $\nabla C^{abs}$ | **0.5050** | 0.6838 | 0.9089 | 0.7334 | 0.8328 | 0.8394 | 0.9288 | 0.8725 |
| | $\mathrm{ACP}^{abs}_{0.5}$ | **0.5116** | 0.6937 | 0.9122 | 0.7334 | 0.8328 | 0.8361 | 0.9288 | 0.8725 |
| | $\mathrm{ACP}^{abs}_{0.8}$ | **0.4255** | 0.5281 | 0.8692 | 0.6440 | 0.7798 | 0.6805 | 0.8890 | 0.7963 |
| ViT | $\mathrm{AUBC}^{rel}$ | **0.8579** | 0.9330 | 0.9897 | 0.9869 | 0.9880 | 0.9856 | 0.9868 | 0.9887 |
| | $\nabla C^{rel}$ | **0.8725** | 0.9520 | 0.9950 | 0.9950 | 0.9950 | 0.9950 | 0.9950 | 0.9950 |
| | $\mathrm{ACP}^{rel}_{0.5}$ | **0.8725** | 0.9520 | – | – | – | – | – | – |
| | $\mathrm{ACP}^{rel}_{0.8}$ | **0.8261** | 0.8890 | – | – | – | – | – | – |
| | $\mathrm{AUBC}^{abs}$ | **0.8967** | 0.9349 | 0.9629 | 0.9307 | 0.9523 | 0.9591 | 0.9685 | 0.9474 |
| | $\nabla C^{abs}$ | **0.9023** | 0.9453 | 0.9685 | 0.9387 | 0.9619 | 0.9685 | 0.9751 | 0.9553 |
| | $\mathrm{ACP}^{abs}_{0.5}$ | **0.9056** | 0.9453 | 0.9718 | 0.9420 | 0.9619 | 0.9685 | 0.9784 | 0.9553 |
| | $\mathrm{ACP}^{abs}_{0.8}$ | **0.8791** | 0.9222 | 0.9619 | 0.9321 | 0.9453 | 0.9486 | 0.9652 | 0.9453 |

Table 7: ImageNet-S: Quantitative robustness evaluation across models, datasets, and attribution methods with **Black Perturbation** and **cell size of 14**. Dashes (**–**) indicate that the confidence threshold was not reached

| Model | Metric | (A) Random | (B) SHAP | (C) GC | (D) SC | (E) IG | (F) AM | (G) OS | (H) Loss |
|---|---|---|---|---|---|---|---|---|---|
| ResNet34 | $\mathrm{AUBC}^{rel}$ | **0.4308** | 0.6842 | 0.7327 | 0.7751 | 0.8274 | 0.8223 | 0.9222 | 0.9005 |
| | $\nabla C^{rel}$ | **0.4387** | 0.6904 | 0.7202 | 0.8063 | 0.8361 | 0.8361 | 0.9486 | 0.9288 |
| | $\mathrm{ACP}_{0.5}^{rel}$ | **0.4387** | 0.7003 | 0.7566 | 0.8228 | 0.8394 | 0.8361 | 0.9520 | 0.9288 |
| | $\mathrm{ACP}_{0.8}^{rel}$ | **0.3195** | 0.4884 | 0.5116 | 0.5778 | 0.7698 | 0.7334 | 0.8791 | 0.8559 |
| | $\mathrm{AUBC}^{abs}$ | **0.5722** | 0.6180 | 0.6643 | 0.7033 | 0.7596 | 0.7756 | 0.8067 | 0.8544 |
| | $\nabla C^{abs}$ | **0.5712** | 0.6175 | 0.6606 | 0.7036 | 0.7698 | 0.7831 | 0.8129 | 0.8692 |
| | $\mathrm{ACP}_{0.5}^{abs}$ | **0.5745** | 0.6209 | 0.6639 | 0.7069 | 0.7698 | 0.7864 | 0.8162 | 0.8725 |
| | $\mathrm{ACP}_{0.8}^{abs}$ | **0.4487** | 0.5083 | 0.5778 | 0.6142 | 0.6937 | 0.6672 | 0.7103 | 0.7996 |
| ResNet50 | $\mathrm{AUBC}^{rel}$ | **0.3643** | 0.5917 | 0.8580 | 0.7787 | 0.8618 | 0.8165 | 0.9754 | 0.9431 |
| | $\nabla C^{rel}$ | **0.3692** | 0.5811 | 0.9420 | 0.8162 | 0.8725 | 0.8361 | 0.9950 | 0.9718 |
| | $\mathrm{ACP}_{0.5}^{rel}$ | **0.3692** | 0.5877 | 0.9553 | 0.8427 | 0.8758 | 0.8361 | – | 0.9718 |
| | $\mathrm{ACP}_{0.8}^{rel}$ | **0.2897** | 0.4387 | 0.7069 | 0.5679 | 0.8460 | 0.7301 | 0.9652 | 0.9387 |
| | $\mathrm{AUBC}^{abs}$ | 0.6061 | **0.6044** | 0.7685 | 0.7131 | 0.8049 | 0.7512 | 0.8871 | 0.8694 |
| | $\nabla C^{abs}$ | **0.5646** | 0.5911 | 0.7698 | 0.7136 | 0.8162 | 0.7632 | 0.8990 | 0.8890 |
| | $\mathrm{ACP}_{0.5}^{abs}$ | **0.5944** | 0.6010 | 0.7732 | 0.7202 | 0.8162 | 0.7632 | 0.8990 | 0.8890 |
| | $\mathrm{ACP}_{0.8}^{abs}$ | **0.4089** | 0.4652 | 0.7036 | 0.5745 | 0.7798 | 0.6871 | 0.8824 | 0.8228 |
| VGG16 | $\mathrm{AUBC}^{rel}$ | **0.5329** | 0.5508 | 0.7351 | 0.6550 | 0.8514 | 0.6965 | 0.8920 | 0.7862 |
| | $\nabla C^{rel}$ | **0.5315** | 0.5414 | 0.7599 | 0.6473 | 0.9056 | 0.7036 | 0.9056 | 0.8096 |
| | $\mathrm{ACP}_{0.5}^{rel}$ | **0.5348** | 0.5480 | 0.7632 | 0.6573 | 0.9023 | 0.7103 | 0.9056 | 0.8096 |
| | $\mathrm{ACP}_{0.8}^{rel}$ | 0.4652 | **0.3891** | 0.5712 | 0.4851 | 0.7599 | 0.5182 | 0.8328 | 0.6937 |
| | $\mathrm{AUBC}^{abs}$ | **0.5244** | 0.5794 | 0.7209 | 0.6222 | 0.7024 | 0.6613 | 0.8139 | 0.7839 |
| | $\nabla C^{abs}$ | **0.5248** | 0.5778 | 0.7500 | 0.6209 | 0.7069 | 0.6473 | 0.8195 | 0.8030 |
| | $\mathrm{ACP}_{0.5}^{abs}$ | **0.5281** | 0.5811 | 0.7467 | 0.6242 | 0.7069 | 0.6606 | 0.8228 | 0.8030 |
| | $\mathrm{ACP}_{0.8}^{abs}$ | **0.4553** | 0.4553 | 0.6010 | 0.5017 | 0.6242 | 0.4818 | 0.7665 | 0.6904 |
| EfficientNet | $\mathrm{AUBC}^{rel}$ | **0.7585** | 0.7696 | 0.7907 | 0.8212 | 0.8971 | 0.9037 | 0.9066 | 0.8847 |
| | $\nabla C^{rel}$ | **0.7732** | 0.8294 | 0.9155 | 0.8659 | 0.9486 | 0.9586 | 0.9884 | 0.9188 |
| | $\mathrm{ACP}_{0.5}^{rel}$ | **0.7765** | 0.8228 | 0.8957 | 0.8692 | 0.9486 | 0.9586 | 0.9851 | 0.9188 |
| | $\mathrm{ACP}_{0.8}^{rel}$ | 0.6573 | **0.6341** | 0.6540 | 0.6738 | 0.8294 | 0.9122 | 0.8890 | 0.8692 |
| | $\mathrm{AUBC}^{abs}$ | 0.8373 | **0.7551** | 0.8345 | 0.7740 | 0.8245 | 0.8630 | 0.8597 | 0.8586 |
| | $\nabla C^{abs}$ | 0.8659 | **0.7831** | 0.8990 | 0.7897 | 0.8394 | 0.8990 | 0.9056 | 0.8791 |
| | $\mathrm{ACP}_{0.5}^{abs}$ | 0.8659 | **0.7831** | 0.8957 | 0.7897 | 0.8394 | 0.8990 | 0.9056 | 0.8791 |
| | $\mathrm{ACP}_{0.8}^{abs}$ | 0.7500 | **0.6606** | 0.7169 | 0.7103 | 0.7698 | 0.8030 | 0.8361 | 0.8162 |
| DenseNet | $\mathrm{AUBC}^{rel}$ | **0.6507** | 0.6964 | 0.9836 | 0.7696 | 0.9306 | 0.8607 | 0.9679 | 0.9616 |
| | $\nabla C^{rel}$ | **0.6440** | 0.6970 | 0.9950 | 0.7864 | 0.9420 | 0.8824 | 0.9950 | 0.9950 |
| | $\mathrm{ACP}_{0.5}^{rel}$ | **0.6507** | 0.7268 | – | 0.7864 | 0.9420 | 0.8857 | – | – |
| | $\mathrm{ACP}_{0.8}^{rel}$ | 0.5546 | **0.4652** | 0.9950 | 0.6805 | 0.9122 | 0.8559 | 0.9520 | 0.9387 |
| | $\mathrm{AUBC}^{abs}$ | **0.6190** | 0.7169 | 0.8977 | 0.7407 | 0.8318 | 0.7713 | 0.8988 | 0.8570 |
| | $\nabla C^{abs}$ | **0.6209** | 0.7235 | 0.9089 | 0.7467 | 0.8361 | 0.7698 | 0.9122 | 0.8592 |
| | $\mathrm{ACP}_{0.5}^{abs}$ | **0.6209** | 0.7268 | 0.9089 | 0.7467 | 0.8394 | 0.7732 | 0.9122 | 0.8626 |
| | $\mathrm{ACP}_{0.8}^{abs}$ | **0.5679** | 0.5778 | 0.8824 | 0.6738 | 0.7831 | 0.6805 | 0.8526 | 0.8162 |
| ViT | $\mathrm{AUBC}^{rel}$ | **0.9013** | 0.9311 | 0.9897 | 0.9784 | 0.9882 | 0.9853 | 0.9884 | 0.9891 |
| | $\nabla C^{rel}$ | **0.9155** | 0.9520 | 0.9950 | 0.9851 | 0.9950 | 0.9950 | 0.9950 | 0.9950 |
| | $\mathrm{ACP}_{0.5}^{rel}$ | **0.9188** | 0.9553 | – | 0.9884 | – | – | – | – |
| | $\mathrm{ACP}_{0.8}^{rel}$ | **0.8659** | 0.8824 | – | 0.9818 | – | – | – | – |
| | $\mathrm{AUBC}^{abs}$ | 0.9358 | **0.9047** | 0.9691 | 0.9290 | 0.9589 | 0.9579 | 0.9780 | 0.9465 |
| | $\nabla C^{abs}$ | 0.9486 | **0.9122** | 0.9751 | 0.9354 | 0.9685 | 0.9685 | 0.9851 | 0.9520 |
| | $\mathrm{ACP}_{0.5}^{abs}$ | 0.9486 | **0.9155** | 0.9784 | 0.9354 | 0.9685 | 0.9685 | 0.9884 | 0.9553 |
| | $\mathrm{ACP}_{0.8}^{abs}$ | 0.9321 | 0.8659 | 0.9685 | 0.9222 | 0.9619 | 0.9453 | 0.9851 | 0.9453 |

Table 8: ImageNet-S: Quantitative robustness evaluation across models, datasets, and attribution methods with **White Perturbation** and **cell size of 14**. Dashes (**–**) indicate that the confidence threshold was not reached

| Model | Metric | (A) Random | (B) SHAP | (C) GC | (D) SC | (E) IG | (F) AM | (G) OS | (H) Loss |
|---|---|---|---|---|---|---|---|---|---|
| ResNet34 | $\text{AUBC}^{rel}$ | 0.9900 | 0.9900 | 0.9900 | 0.9900 | 0.9900 | 0.9900 | 0.9900 | 0.9900 |
| | $\nabla C^{rel}$ | 0.5579 | 0.7599 | 0.0646 | 0.0480 | 0.0480 | 0.0480 | 0.0845 | 0.2699 |
| | $\text{ACP}^{rel}_{0.5}$ | – | – | – | – | – | – | – | – |
| | $\text{ACP}^{rel}_{0.8}$ | – | – | – | – | – | – | – | – |
| | $\text{AUBC}^{abs}$ | 0.9900 | 0.9900 | 0.9900 | 0.9900 | 0.9900 | 0.9900 | 0.9900 | 0.9900 |
| | $\nabla C^{abs}$ | 0.9222 | 0.5116 | 0.5944 | 0.0083 | 0.0812 | 0.7036 | 0.5679 | 0.7864 |
| | $\text{ACP}^{abs}_{0.5}$ | – | – | – | – | – | – | – | – |
| | $\text{ACP}^{abs}_{0.8}$ | – | – | – | – | – | – | – | – |
| ResNet50 | $\text{AUBC}^{rel}$ | 0.9900 | 0.9900 | 0.9900 | 0.9900 | 0.9900 | 0.9900 | 0.9900 | 0.9900 |
| | $\nabla C^{rel}$ | 0.0249 | 0.0216 | 0.5116 | 0.4520 | 0.0778 | 0.0547 | 0.3493 | 0.0050 |
| | $\text{ACP}^{rel}_{0.5}$ | – | – | – | – | – | – | – | – |
| | $\text{ACP}^{rel}_{0.8}$ | – | – | – | – | – | – | – | – |
| | $\text{AUBC}^{abs}$ | 0.9900 | 0.9900 | 0.9900 | 0.9900 | 0.9900 | 0.9900 | 0.9900 | 0.9900 |
| | $\nabla C^{abs}$ | 0.7036 | 0.8957 | 0.9950 | 0.4321 | 0.1871 | 0.0050 | 0.0480 | 0.1606 |
| | $\text{ACP}^{abs}_{0.5}$ | – | – | – | – | – | – | – | – |
| | $\text{ACP}^{abs}_{0.8}$ | – | – | – | – | – | – | – | – |
| VGG16 | $\text{AUBC}^{rel}$ | 0.9900 | 0.9900 | 0.9900 | 0.9900 | 0.9900 | 0.9900 | 0.9900 | 0.9900 |
| | $\nabla C^{rel}$ | 0.2004 | 0.0050 | 0.9652 | 0.2467 | 0.0050 | 0.4652 | 0.0050 | 0.5414 |
| | $\text{ACP}^{rel}_{0.5}$ | – | – | – | – | – | – | – | – |
| | $\text{ACP}^{rel}_{0.8}$ | – | – | – | – | – | – | – | – |
| | $\text{AUBC}^{abs}$ | 0.9900 | 0.9900 | 0.9900 | 0.9900 | 0.9900 | 0.9900 | 0.9900 | 0.9900 |
| | $\nabla C^{abs}$ | 0.9950 | 0.6209 | 0.9950 | 0.9950 | 0.7864 | 0.7301 | 0.0447 | 0.1374 |
| | $\text{ACP}^{abs}_{0.5}$ | – | – | – | – | – | – | – | – |
| | $\text{ACP}^{abs}_{0.8}$ | – | – | – | – | – | – | – | – |
| EfficientNet | $\text{AUBC}^{rel}$ | 0.9851 | 0.9860 | 0.9855 | 0.9876 | 0.9878 | 0.9866 | 0.9865 | 0.9864 |
| | $\nabla C^{rel}$ | 0.0050 | 0.0050 | 0.0050 | 0.9950 | 0.9950 | 0.0050 | 0.9950 | 0.0050 |
| | $\text{ACP}^{rel}_{0.5}$ | – | – | – | – | – | – | – | – |
| | $\text{ACP}^{rel}_{0.8}$ | – | – | – | – | – | – | – | – |
| | $\text{AUBC}^{abs}$ | 0.9873 | 0.9872 | 0.9877 | 0.9885 | 0.9888 | 0.9877 | 0.9875 | 0.9875 |
| | $\nabla C^{abs}$ | 0.0050 | 0.0050 | 0.0050 | 0.9950 | 0.9950 | 0.0050 | 0.9950 | 0.0050 |
| | $\text{ACP}^{abs}_{0.5}$ | – | – | – | – | – | – | – | – |
| | $\text{ACP}^{abs}_{0.8}$ | – | – | – | – | – | – | – | – |
| DenseNet | $\text{AUBC}^{rel}$ | 0.9900 | 0.9900 | 0.9900 | 0.9900 | 0.9900 | 0.9900 | 0.9900 | 0.9900 |
| | $\nabla C^{rel}$ | 0.1010 | 0.1143 | 0.0381 | 0.0050 | 0.9089 | 0.0414 | 0.5381 | 0.0249 |
| | $\text{ACP}^{rel}_{0.5}$ | – | – | – | – | – | – | – | – |
| | $\text{ACP}^{rel}_{0.8}$ | – | – | – | – | – | – | – | – |
| | $\text{AUBC}^{abs}$ | 0.9900 | 0.9900 | 0.9900 | 0.9900 | 0.9900 | 0.9900 | 0.9900 | 0.9900 |
| | $\nabla C^{abs}$ | 0.1772 | 0.2202 | 0.5944 | 0.0778 | 0.0050 | 0.0182 | 0.5116 | 0.8692 |
| | $\text{ACP}^{abs}_{0.5}$ | – | – | – | – | – | – | – | – |
| | $\text{ACP}^{abs}_{0.8}$ | – | – | – | – | – | – | – | – |
| ViT | $\text{AUBC}^{rel}$ | 0.9898 | 0.9898 | 0.9898 | 0.9898 | 0.9898 | 0.9898 | 0.9898 | 0.9898 |
| | $\nabla C^{rel}$ | 0.5414 | 0.0050 | 0.0149 | 0.0646 | 0.9751 | 0.9950 | 0.5712 | 0.0050 |
| | $\text{ACP}^{rel}_{0.5}$ | – | – | – | – | – | – | – | – |
| | $\text{ACP}^{rel}_{0.8}$ | – | – | – | – | – | – | – | – |
| | $\text{AUBC}^{abs}$ | 0.9898 | 0.9898 | 0.9898 | 0.9898 | 0.9898 | 0.9898 | 0.9898 | 0.9898 |
| | $\nabla C^{abs}$ | 0.6904 | 0.9453 | 0.9917 | – | 0.5646 | 0.9188 | 0.8626 | 0.9586 |
| | $\text{ACP}^{abs}_{0.5}$ | – | – | – | – | – | – | – | – |
| | $\text{ACP}^{abs}_{0.8}$ | – | – | – | – | – | – | – | – |

Table 9: ImageNet-S: Quantitative robustness evaluation across models, datasets, and attribution methods with **Blurr Perturbation** and **cell size of 14**. Dashes (**–**) indicate that the confidence threshold was not reached

| Perturbation | Metric | (A) Random | (B) SHAP | (C) GC | (D) SC | (E) IG | (F) AM | (G) OS | (H) Loss |
|---|---|---|---|---|---|---|---|---|---|
| MDP | $\text{AUBC}^{rel}$ | **0.5871** | 0.6259 | 0.6805 | 0.6192 | 0.6676 | 0.6415 | 0.8616 | 0.6959 |
| | $\nabla C^{rel}$ | **0.5447** | 0.6076 | 0.6805 | 0.6043 | 0.6639 | 0.6308 | 0.9056 | 0.6937 |
| | $\text{ACP}^{rel}_{0.5}$ | **0.5745** | 0.6242 | 0.6871 | 0.6142 | 0.6705 | 0.6407 | 0.9056 | 0.6970 |
| | $\text{ACP}^{rel}_{0.8}$ | **0.3858** | 0.4354 | 0.5248 | 0.4553 | 0.5480 | 0.4917 | 0.7831 | 0.5646 |
| | $\text{AUBC}^{abs}$ | **0.5785** | 0.6053 | 0.6881 | 0.6050 | 0.6660 | 0.6257 | 0.7779 | 0.6708 |
| | $\nabla C^{abs}$ | **0.5281** | 0.5811 | 0.6871 | 0.5911 | 0.6639 | 0.6109 | 0.8030 | 0.6639 |
| | $\text{ACP}^{abs}_{0.5}$ | **0.5646** | 0.6010 | 0.6937 | 0.6010 | 0.6672 | 0.6209 | 0.8030 | 0.6705 |
| | $\text{ACP}^{abs}_{0.8}$ | **0.3725** | 0.4156 | 0.5480 | 0.4520 | 0.5546 | 0.4685 | 0.6805 | 0.5315 |
| Black | $\text{AUBC}^{rel}$ | **0.6194** | 0.7143 | 0.7143 | 0.6577 | 0.7432 | 0.6870 | 0.9318 | 0.7910 |
| | $\nabla C^{rel}$ | **0.5877** | 0.7202 | 0.7169 | 0.6573 | 0.7467 | 0.6904 | 0.9652 | 0.8030 |
| | $\text{ACP}^{rel}_{0.5}$ | **0.6109** | 0.7301 | 0.7235 | 0.6639 | 0.7467 | 0.6937 | 0.9652 | 0.8030 |
| | $\text{ACP}^{rel}_{0.8}$ | **0.4288** | 0.5149 | 0.5745 | 0.5281 | 0.6507 | 0.5646 | 0.9188 | 0.6937 |
| | $\text{AUBC}^{abs}$ | **0.6131** | 0.6985 | 0.7155 | 0.6616 | 0.7394 | 0.7023 | 0.8754 | 0.7820 |
| | $\nabla C^{abs}$ | **0.5844** | 0.7069 | 0.7235 | 0.6639 | 0.7401 | 0.7069 | 0.9023 | 0.7930 |
| | $\text{ACP}^{abs}_{0.5}$ | **0.6043** | 0.7136 | 0.7235 | 0.6672 | 0.7434 | 0.7103 | 0.9023 | 0.7963 |
| | $\text{ACP}^{abs}_{0.8}$ | **0.4255** | 0.5215 | 0.5944 | 0.5315 | 0.6507 | 0.5679 | 0.8361 | 0.6738 |
| White | $\text{AUBC}^{rel}$ | 0.6927 | **0.6638** | 0.7046 | 0.6635 | 0.7355 | 0.6791 | 0.9399 | 0.8150 |
| | $\nabla C^{rel}$ | 0.6771 | **0.6573** | 0.7003 | 0.6606 | 0.7367 | 0.6771 | 0.9851 | 0.8294 |
| | $\text{ACP}^{rel}_{0.5}$ | 0.6904 | **0.6672** | 0.7069 | 0.6639 | 0.7401 | 0.6805 | 0.9851 | 0.8294 |
| | $\text{ACP}^{rel}_{0.8}$ | 0.5381 | **0.4851** | 0.5579 | 0.5248 | 0.6374 | 0.5513 | 0.9122 | 0.7202 |
| | $\text{AUBC}^{abs}$ | 0.6825 | **0.6563** | 0.7212 | 0.6714 | 0.7489 | 0.7167 | 0.8810 | 0.8059 |
| | $\nabla C^{abs}$ | 0.6672 | **0.6440** | 0.7202 | 0.6672 | 0.7500 | 0.7169 | 0.9089 | 0.8195 |
| | $\text{ACP}^{abs}_{0.5}$ | 0.6771 | **0.6573** | 0.7268 | 0.6738 | 0.7533 | 0.7235 | 0.9089 | 0.8195 |
| | $\text{ACP}^{abs}_{0.8}$ | 0.5182 | **0.4752** | 0.5911 | 0.5348 | 0.6473 | 0.5712 | 0.8294 | 0.7003 |

Table 10: Oxford-Flowers ResNet50: Quantitative robustness evaluation across models, datasets, and attribution methods with **cell size of 14**. Dashes (–) indicate that the confidence threshold was not reached

| Perturbation | Metric | (A) Random | (B) SHAP | (C) GC | (D) SC | (E) IG | (F) AM | (G) OS | (H) Loss |
|---|---|---|---|---|---|---|---|---|---|
| MDP | $\text{AUBC}^{rel}$ | **0.5555** | 0.7414 | 0.9184 | 0.7913 | 0.9508 | 0.9002 | 0.8965 | 0.8932 |
| | $\nabla C^{rel}$ | **0.4884** | 0.7599 | 0.9950 | 0.7963 | 0.9784 | 0.9222 | 0.9619 | 0.9023 |
| | $\text{ACP}_{0.5}^{rel}$ | **0.5381** | 0.7632 | – | 0.7996 | 0.9784 | 0.9255 | 0.9586 | 0.9023 |
| | $\text{ACP}_{0.8}^{rel}$ | **0.3361** | 0.5811 | 0.8526 | 0.6871 | 0.9255 | 0.8294 | 0.8626 | 0.8361 |
| | $\text{AUBC}^{abs}$ | 0.8060 | **0.7889** | 0.8703 | 0.7985 | 0.8802 | 0.8589 | 0.8519 | 0.8518 |
| | $\nabla C^{abs}$ | 0.8129 | **0.7996** | 0.8824 | 0.8063 | 0.8890 | 0.8692 | 0.8957 | 0.8559 |
| | $\text{ACP}_{0.5}^{abs}$ | 0.8162 | **0.8030** | 0.8857 | 0.8096 | 0.8890 | 0.8725 | 0.8924 | 0.8592 |
| | $\text{ACP}_{0.8}^{abs}$ | 0.7235 | **0.6805** | 0.8030 | 0.7003 | 0.8328 | 0.7798 | 0.8030 | 0.7831 |
| Black | $\text{AUBC}^{rel}$ | **0.6006** | 0.7194 | 0.9096 | 0.7994 | 0.9491 | 0.9005 | 0.9099 | 0.9096 |
| | $\nabla C^{rel}$ | **0.5546** | 0.7268 | 0.9917 | 0.8096 | 0.9685 | 0.9222 | 0.9685 | 0.9155 |
| | $\text{ACP}_{0.5}^{rel}$ | **0.5911** | 0.7334 | 0.9917 | 0.8129 | 0.9718 | 0.9222 | 0.9652 | 0.9188 |
| | $\text{ACP}_{0.8}^{rel}$ | **0.3891** | 0.5480 | 0.8361 | 0.6838 | 0.9288 | 0.8294 | 0.8990 | 0.8824 |
| | $\text{AUBC}^{abs}$ | 0.8240 | **0.7641** | 0.8697 | 0.8033 | 0.8909 | 0.8701 | 0.8623 | 0.8487 |
| | $\nabla C^{abs}$ | 0.8294 | **0.7698** | 0.8791 | 0.8096 | 0.8990 | 0.8791 | 0.9089 | 0.8526 |
| | $\text{ACP}_{0.5}^{abs}$ | 0.8328 | **0.7732** | 0.8824 | 0.8129 | 0.9023 | 0.8824 | 0.9056 | 0.8559 |
| | $\text{ACP}_{0.8}^{abs}$ | 0.7467 | **0.6341** | 0.8030 | 0.7069 | 0.8427 | 0.8030 | 0.8294 | 0.7864 |
| White | $\text{AUBC}^{rel}$ | **0.5931** | 0.6873 | 0.9045 | 0.7695 | 0.9409 | 0.9088 | 0.8813 | 0.9129 |
| | $\nabla C^{rel}$ | 0.5281 | 0.6871 | 0.9950 | 0.7732 | 0.9718 | 0.9387 | 0.9586 | 0.9222 |
| | $\text{ACP}_{0.5}^{rel}$ | **0.5745** | 0.7003 | – | 0.7798 | 0.9718 | 0.9387 | 0.9553 | 0.9222 |
| | $\text{ACP}_{0.8}^{rel}$ | **0.3692** | 0.4917 | 0.8162 | 0.6407 | 0.9056 | 0.8427 | 0.8261 | 0.8725 |
| | $\text{AUBC}^{abs}$ | 0.7944 | **0.7681** | 0.8724 | 0.8224 | 0.8869 | 0.8741 | 0.8410 | 0.8566 |
| | $\nabla C^{abs}$ | 0.7963 | **0.7831** | 0.8924 | 0.8328 | 0.8957 | 0.8857 | 0.8924 | 0.8626 |
| | $\text{ACP}_{0.5}^{abs}$ | 0.7996 | **0.7864** | 0.8957 | 0.8361 | 0.8957 | 0.8890 | 0.8890 | 0.8626 |
| | $\text{ACP}_{0.8}^{abs}$ | 0.6970 | **0.6308** | 0.7963 | 0.7367 | 0.8361 | 0.8162 | 0.7864 | 0.7963 |

Table 11: Oxford-Pets ViT:: Quantitative robustness evaluation across models, datasets, and attribution methods with **cell size of 14**. Dashes (**–**) indicate that the confidence threshold was not reached

