# OpenReview forum: "How to Break Image Classification Models: Random Noise knows better than Saliency"
_ICLR.cc/2026/Conference — Submitted to ICLR 2026_

### Official Review · Reviewer_PDUi · 2025-10-31

**Soundness:** 3
**Presentation:** 3
**Contribution:** 2
**Rating:** 6
**Confidence:** 4

**Summary:**

This paper introduces a  degradation-based evaluation to understand the robustness of image classification models by progressively removing parts of the input image based on attribution maps or random masking. The core hypothesis is that: If an attribution method accurately identifies important regions, then masking those regions should maximally degrade model predictions. The paper quantifies this with 3 methods: Area under the blindness curve, sensitivity slope, and attribution collapse point. The central finding is that: Random masking often causes stronger prediction collapse than saliency-based masking, implying that attribution methods may not align with true model dependencies.

**Strengths:**

- The paper is quite well written and organized.
- The paper shows pretty consistently that: Random masking outperforms attribution methods under object-relative perturbations.
- The paper demonstrates very comprehensive results across ImageNet-S50, Oxford Flowers-102, Oxford-IIIT Pets, and several model classes. The results presented are resilient and well demonstrated. The list of methods tested is also quite comprehensive: Mean-Distance Perturbation (MDP), black, white, and Gaussian blur (excluded for ineffectiveness). Overall, this work is an impressive empirical undertaking.

**Weaknesses:**

My main feedback is that the finding in this work has been previously demonstrated in the Roar paper by Hooker et. al. that the authors identify. In fact, the reason for this issue has been thoroughly, though under the radar, studied in the literature, and these findings almost entirely resolved. I'll discuss them here.

**Summary**

The empirical observation is not new—it replicates the well-documented failure of attributions on non-robust models (Hooker 2019; Shah 2021; Srinivas 2023). Your main contribution is the clean, reproducible degradation-curve framework for quantifying this behavior. To maximize impact, interpret the results as evidence about model robustness regimes, normalize perturbation budgets, and integrate prior theoretical explanations (feature leakage + off-manifold robustness). With these clarifications, the paper will make a strong and properly contextualized contribution

**Overview of prior literature**\
The main empirical  observation in this paper, i.e., that random masking can outperform attribution-guided masking in degrading classifier performance has been shown in prior work. As the authors cite, Hooker et al. (2019, ROAR) first showed that standard saliency methods (including input gradients, IG, and Guided Backprop) perform no better than random baselines when feature importance is assessed through retraining on masked data.

Shah et al. (2021, DiffROAR, Do input gradients highlight discriminative features) formally tested the same assumption: “features with larger gradients are more discriminative” and demonstrated both empirically and theoretically that standard models violate this assumption due to feature leakage, while adversarially robust models satisfy it. Srinivas et al. (2023, Which Models have Perceptually-Aligned Gradients? An Explanation via Off-Manifold Robustness) extended this line by linking attribution fidelity to off-manifold robustness: models robust to small, non-semantic perturbations exhibit input gradients aligned with the signal manifold, producing faithful saliency.

Together, these works establish that (a) random or uninformative perturbations can appear more “effective” on non-robust models, and (b) attribution quality is fundamentally governed by the model’s robustness regime rather than by the visualization method itself.

To state this more plainly: standard neural networks have very small local lipschitz constants, i.e., it requires only a very small perturbation (e.g. adversarial examples) to completely change the output. Even more, the size and type of perturbation that one uses to test whether a model's output is changing are quite crucial. This means that the finding that random masking beats saliency methods is only true for model trained without regularization and that are not robust to perturbations.


**Overview of prior literature**
Viewed through this lens, your finding that random masking often induces stronger degradation should be understood as a measurement of model brittleness, not as evidence that random perturbations carry more causal information.

Perturbation-budget mismatch: Standard CNNs are highly sensitive, tiny amounts of zeroing-out already collapse predictions, whereas robust or regularized models require a larger budget before comparable degradation occurs. Without normalizing for this budget, random masking naturally seems “better.”

Feature leakage: Saliency maps of non-robust models highlight both genuine and spurious correlations. Masking by those maps therefore removes a mixture of relevant and irrelevant pixels, producing weaker degradation than random masking, which happens to disrupt fragile off-manifold features more uniformly.

Model-dependent regime: In the terminology of Srinivas et al., your experiments probe the weak-robustness regime, where gradients live off-manifold. In a Bayes-aligned regime (adversarially or contrastively robust models), attribution-guided masking should surpass random baselines.

Thus, the correct takeaway is that your degradation metrics—AUBC, sensitivity slope, and ACP—quantify robustness to input removal, not explanatory fidelity. Your framework is valuable precisely because it exposes this dependence.

**Questions:**

How to improve the work:

**Reposition the contribution.**
- State explicitly that the novelty lies in a behavioral robustness framework for quantifying model fragility under input removal, rather than in rediscovering that random > saliency.

**Control for perturbation budget.**
- Normalize comparisons by matching an equal initial confidence drop or expected loss change, or report degradation curves in “equivalent-confidence” space. This will disentangle attribution quality from model sensitivity.

**Connect to existing theory.**
- Cite and discuss how your empirical findings align with the feature-leakage analysis of Shah et al. (2021) and the off-manifold robustness theory of Srinivas et al. (2023). Clarify that your framework provides an operational test for these effects.

**Extend experiments.**
- Evaluate degradation on robustly trained models or Vision Transformers to show that in the Bayes-aligned regime, attribution-guided masking indeed overtakes random.

- Optionally, cross-validate with DiffROAR metrics to confirm that negative DiffROAR scores coincide with the random > saliency regime.

**Interpret results accordingly.**
Emphasize that random outperforming saliency is diagnostic of non-robust feature reliance. When attribution finally exceeds random under stronger robustness, that transition pinpoints where models begin to rely on semantically meaningful evidence.

---

### Official Review · Reviewer_GcuP · 2025-11-01

**Soundness:** 2
**Presentation:** 2
**Contribution:** 1
**Rating:** 2
**Confidence:** 3

**Summary:**

The paper addresses the problem of understanding how vulnerable models' decisions are to the removal of supposedly important evidence. For this purpose, the authors introduce a framework to derive three quantitative metrics: Area Under the Blindness Curve (AUBC), Sensitivity Slope, and Attribution Collapse Point. The authors claim to reveal a fundamental gap between visual explanations and true decision dependencies, using empirical evidence across standard architectures, and found that random masking often surpasses attribution-based masking.

**Strengths:**

1) The paper explores an important problem, particularly for safety-critical applications in XAI i.e., the robustness of classifier predictions to evidence removal.

2) The authors present a coherent, formalized framework that generalizes degradation-based evaluation of explanations.

3) The method can be applied to any black-box image classifier and multiple explanation methods (Grad-CAM, SHAP, etc.).

**Weaknesses:**

1) The literature survey seems limited as the paper does not mention Zheng et. al. (F-Fidelity: A Robust Framework for Faithfulness Evaluation of Explainable AI, ICLR 2024), which was an important paper in this direction.

2) The authors report that random masking often surpasses attribution-based masking, but did not discuss whether their masking methodology can lead to OOD issues. It is well known from prior works that models' behavior in OOD scenarios differs from that within the distribution.

3) The patch size 14 seems arbitrary, and it would be helpful to readers if the authors provided some justification for choosing the particular patch size.

4) The paper repeatedly refers to “model confidence” as the quantity being tracked along the degradation curve. Could the authors clarify whether this represents a calibrated probability (e.g., after temperature scaling) or the raw softmax output? If the latter, how do the proposed metrics disentangle robustness degradation from potential calibration errors, given that uncalibrated models can exhibit large confidence fluctuations unrelated to actual correctness?

5)  Tomsett et.al. (Sanity Checks for Saliency Metrics, AAAI 2020) reported that well-known fidelity metrics were unreliable. Given this background, it remains unclear how the proposed metrics (AUBC, Sensitivity Slope, ACP) mitigate such reliability issues.

**Questions:**

I request the authors to address the weakness above.

---

### Official Review · Reviewer_K6gc · 2025-11-01

**Soundness:** 3
**Presentation:** 2
**Contribution:** 2
**Rating:** 2
**Confidence:** 3

**Summary:**

This paper introduces a degradation-based evaluation framework based on masking out saliency maps of different attribution methods or masking random pixels.
Based on this framework, they investigate their hypothesis that masking out the most important input regions identified by attribution methods should cause maximal prediction collapse. However, they find that masking out random pixels often results in a greater prediction collapse.
They further introduce three metrics (Area Under the Blindness Curve (AUBC), sensitivity slope, Attribution Collapse Point (ACP)) to analyse the confidence collapse of image classifiers under iterative input masking.

**Strengths:**

The fine-grained analysis of vision model robustness to input masking including their introduced quantitative metrics. Especially the different behaviour of ViTs seems interesting.

**Weaknesses:**

At the beginning, the authors claim: “While saliency methods are widely used to interpret model predictions, their ability to identify functionally critical features remains largely untested”. However,
there are several papers on feature attributions that consider (iterative) input perturbations to evaluate feature attribution methods, e.g. [1], [2], [3], [4]. [4] also discuss the effects of artifacts introduced by masking random pixels and suggest to mask out the least salient pixels instead of the most salient.

It seems to be unclear, to which extent random masking degrades the actual information content of the input image compared to using the saliency maps. To evaluate this aspect, I would suggest to add a figure showing examples of masked inputs for the different approaches at different iterations or conducting a user study as a human baseline.

While the paper is written well overall, the focus is in parts not clear, i.e. is the focus the evaluation of attribution methods or the robustness of models towards input perturbations.

Minor: the citation of AutoAttack corresponds to the wrong paper

In its current form, the paper should be rejected, due to (i) the missing discussion and comparison to similar methods, (ii) unclear influence of random artefacts/lack of qualitative analysis, (iii) unclear focus.

[1] Evaluating the visualization of what a Deep Neural Network has learned, Samek et al, 2015, https://arxiv.org/pdf/1509.06321

[2] Grad-CAM++: Improved Visual Explanations for Deep Convolutional Networks, Chattopadhyay et al, 2018, https://arxiv.org/pdf/1710.11063

[3] Towards better understanding of gradient-based attribution methods for Deep Neural Networks, Ancona et al, 2018, https://arxiv.org/pdf/1711.06104

[4] Full-Gradient Representation for Neural Network Visualization, Srinivas and Fleuret, 2019, https://arxiv.org/pdf/1905.00780

**Questions:**

1. At what point in the iterative masking process would also humans not be able anymore to identify the class?

2. How does your method compare to other (iterative) perturbation-based evaluations of feature attribution methods?

---

### Official Review · Reviewer_y4mp · 2025-11-03

**Soundness:** 2
**Presentation:** 2
**Contribution:** 2
**Rating:** 6
**Confidence:** 3

**Summary:**

Attribution maps are visualization tools to highlight which regions of the images are "more relevant" for the decision made by a machine learning model (e.g. what is more relevant to assign high probability to certain class). A number of algorithms (or definitions) to compute these attention or attribution maps have been proposed according to different heuristics.  Some of them are based on the gradient of the output depending on the input, and some others depend on how the probability of the class (or performance) drops when removing regions of the input. Both approaches follow the same underlying idea, but the details of the definitions make it difficult to predict what will happen if degradations are applied to the regions identified as relevant.

In this work the authors explictly check the (naive?) hypothesis that masking the "most relevant" regions should cause maximal drop in performance. Given an attribution map the authors mask regions in turn according to their attribution (from more to less relevant) and check the drop in performance. They define different metrics to describe the drop, and they compare the drop with what you get by masking regions in random order. They do this in an absolute way (for the whole image) or in an object-relative way (i.e. removing parts of a selected object). This procedure (except for the description metrics and the size of the regions) is similar to the "deletion/insertion" score proposed by Petsiuk et al. 18.
They find that in object-relative masking (when objects are known in advance and certain object is attacked) random order is more effective (in the considered databases/algorithms) than the order given by attribution algorithms. When considering absolute masking the advantage of random masking is not that evident but still overperforms many attribution methods.

[Despite the authors do not phrase it like this] (to me) these results suggest that the regions identified by the considered visualization tools are not that relevant for the models to make the decisions because randomly chosen regions hurt performance in equivalent or bigger ways. (I think) this implies that the details to get operational definitions of atributtion maps matter, and that these maps should not be interpreted as "regions taken into account by the model to make the decision".

**Strengths:**

This work is an interesting empirical example of the warnings by Jacovi and Goldberg 20 on the disconnection between "functional relevance" and (subjective) "visual plausibility" of the attribution maps.

Even though the authors don't want to phrase it like that (e.g. disclaimers made in line 46-47, 52 of the introduction and lines 460-461 in the conclusion), I think the results do question the quantitative/functional usefulness of the attribution maps: if they do not describe sensible/vulnerable regions of the input (or regions used to make the decision from), what are they useful for?. In short, some implementations of the (same) "attribution" idea seem better than others when it comes to describe the relevance of the identified regions in terms of the vulnerability of the performance depending on the regions.

**Weaknesses:**

(A) The major weakness is the similarity between the proposed test of the relevance of the regions by the drop in performance to the "deletion/insertion" score proposed by Petsiuk et al. 18 (except for the description metrics and the size of the regions). In which way Figs. 2 and 3 of this work are qualitatively different from Fig.2 of Petsiuk et al.?

(B) The object-relative masking should be clarified. Masking from zero to one implies "pre-defining" the region occupied by an object (are the objects segmented by hand?) and then compute the percentage of area deleted from the object?-. In any case, the relative measure (assuming object locations are known and exploited) bias the result in favour of random sampling, since it is not random, but located in a pre-defined object, right?.

(C) Paragraph 394-401 where table 1 is analyzed is not clear. Table displays ACP_0.8^abs as the worse descriptor for random sampling while the paragraph talks about ACP_0.5^abs and AUBC^abs (which seem fine in the table for random sampling).

(D) Authors find that ViT are the more robust (of the considered architectures) to different kinds of masking. Have the authors compared the attention regions of ViT with the attribution regions given by the considered attribution methods?

(E) If the second paragraph of what I find as a "strength" of the work is true, the authors could elaborate on comparison between the proposed method to "evaluate" attribution algorithms and alternative evaluations already proposed before.

(F) As said in the "strengths" section above, I think the authors dont want to criticise much attribution methods. Could the authors elaborate on the fact that if attribution regions do not describe sensible/vulnerable regions of the input (or regions used to make the decision from), what are they useful for? [I did not know this visualization literature and I found it -qualitatively- interesting but -quantitatively- questioned by the presented results].

**Questions:**

Please see questions (A)-(F) in the weaknesses section.

---

### Meta-Review · Area_Chair_tice · 2026-01-09

**Summary:**

Across reviewers, the central concern was limited novelty and insufficient positioning relative to closely related perturbation-based attribution evaluation work (e.g., deletion/insertion, ROAR/DiffROAR-style analyses, and recent faithfulness frameworks). While the empirical study is broad and the degradation-curve metrics are clearly presented, multiple reviewers felt the work repackages known findings (random masking outperforming attribution-guided masking) without clearly articulating what is fundamentally new beyond metric formalization. Additional weaknesses included insufficient discussion of masking-induced artifacts / OOD effects, unclear framing (robustness evaluation vs attribution evaluation), and several methodological clarifications (object-relative masking assumptions, patch size justification, calibration of “confidence,” and interpretation of ViT results). Taken together, the committee judged that the paper’s contribution and framing were not strong enough for acceptance in its current form.

**Reviewer Concerns:**

The rebuttal appears to have helped clarify some presentation-level issues (e.g., definitions around metrics and the intended interpretation of results), and may have partially mitigated confusion about the object-relative setting. However, the key issues remain largely outstanding: reviewers were not convinced the work sufficiently differentiates itself from prior perturbation-based evaluations, nor does it fully engage with the strongest existing explanations of the phenomenon (robustness regime, feature leakage, off-manifold effects). In addition, concerns about random masking artifacts and OOD behavior, missing qualitative examples of degradations, and unclear interpretation of confidence collapse versus calibration were not resolved to a degree that changed the perceived core contribution. As a result, while the rebuttal improved clarity, it did not overcome the main novelty/framing deficits motivating rejection.

**Reviewer Scores:**

Reviewer y4mp (6 → likely 5–6): This reviewer was broadly positive about the empirical message but raised substantial novelty/positioning concerns (esp. similarity to deletion/insertion). With fuller discussion, they might slightly lower confidence in acceptability (toward a weak accept → borderline), but likely remain near 5–6 given their interest in the qualitative takeaway. Reviewer K6gc (2 → likely 2–3): This reviewer recommended rejection and emphasized missing comparisons to established perturbation-based evaluation literature and unclear impact of random masking artifacts; discussion might soften the stance slightly if clarifications were strong, but would likely remain reject. Reviewer GcuP (2 → likely 2–3): Their core concerns (missing key related work such as F-Fidelity, OOD issues from masking, patch-size arbitrariness, confidence/calibration ambiguity) appear structural; discussion could raise the score marginally if rebuttal addressed details, but likely stays reject. Reviewer PDUi (6 → likely 5–6): While appreciative of the empirical effort, this reviewer strongly argued the main observation is known and the real value requires reframing around robustness regimes and prior theory; discussion may have pushed them slightly downward (toward 5) unless the rebuttal convincingly repositioned the paper, but they would likely remain borderline rather than firm reject.

---

### Decision · Program_Chairs · 2026-01-26

Reject